# CORL: Research-oriented Deep Offline Reinforcement Learning Library

**Denis Tarasov**
Tinkoff
den.tarasov@tinkoff.ai

**Alexander Nikulin**
Tinkoff
a.p.nikulin@tinkoff.ai

**Dmitry Akimov**
Tinkoff
d.akimov@tinkoff.ai

**Vladislav Kurenkov**
Tinkoff
v.kurenkov@tinkoff.ai

**Sergey Kolesnikov**
Tinkoff
s.s.kolesnikov@tinkoff.ai

## Abstract

CORL[1] is an open-source library that provides thoroughly benchmarked single-file implementations of both deep offline and offline-to-online reinforcement learning algorithms. It emphasizes a simple developing experience with a straightforward codebase and a modern analysis tracking tool. In CORL, we isolate methods implementation into separate single files, making performance-relevant details easier to recognize. Additionally, an experiment tracking feature is available to help log metrics, hyperparameters, dependencies, and more to the cloud. Finally, we have ensured the reliability of the implementations by benchmarking commonly employed D4RL datasets providing a transparent source of results that can be reused for robust evaluation tools such as performance profiles, probability of improvement, or expected online performance.

## 1   Introduction

Deep Offline Reinforcement Learning (Levine et al., 2020) has been showing significant advancements in numerous domains such as robotics (Smith et al., 2022; Kumar et al., 2021), autonomous driving (Diehl et al., 2021) and recommender systems (Chen et al., 2022). Due to such rapid development, many open-source offline RL solutions[2] emerged to help RL practitioners understand and improve well-known offline RL techniques in different fields. On the one hand, they introduce offline RL algorithms standard interfaces and user-friendly APIs, simplifying offline RL methods incorporation into *existing* projects. On the other hand, introduced abstractions may hinder the learning curve for newcomers and the ease of adoption for researchers interested in developing *new* algorithms. One needs to understand the modularity design (several files on average), which (1) can be comprised of thousands of lines of code or (2) can hardly fit for a novel method[3].

In this technical report, we take a different perspective on an offline RL library and also incorporate emerging interest in the offline-to-online setup. We propose CORL (Clean Offline Reinforcement Learning) – minimalistic and isolated single-file implementations of deep offline and offline-to-online RL algorithms, supported by open-sourced D4RL (Fu et al., 2020) benchmark results. The uncomplicated design allows practitioners to read and understand the implementations of the algorithms straightforwardly. Moreover, CORL supports optional integration with experiments tracking tools such as Weighs&Biases (Biewald, 2020), providing practitioners with a convenient way to analyze

---

[1] CORL Repository: `https://github.com/corl-team/CORL`
[2] `https://github.com/hanjuku-kaso/awesome-offline-rl#oss`
[3] `https://github.com/takuseno/d3rlpy/issues/141`

37th Conference on Neural Information Processing Systems (NeurIPS 2023) Track on Datasets and Benchmarks.

the results and behavior of all algorithms, not merely relying on a final performance commonly reported in papers.

We hope that the CORL library will help offline RL newcomers study implemented algorithms and aid the researchers in quickly modifying existing methods without fighting through different levels of abstraction. Finally, the obtained results may serve as a reference point for D4RL benchmarks avoiding the need to re-implement and tune existing algorithms' hyperparameters.

Yaml Configuration File      Single-File Implementation      Experiment Tracking Log

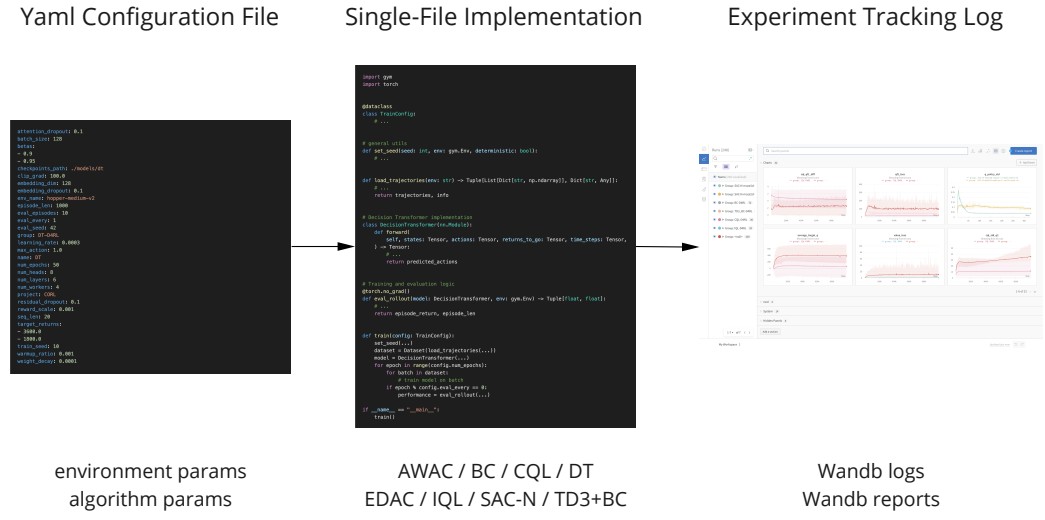

environment params      AWAC / BC / CQL / DT      Wandb logs
algorithm params      EDAC / IQL / SAC-N / TD3+BC      Wandb reports

python dt.py --config=cfg/dt-hopper.yaml --logdir=logs/dt-hopper --num-epochs=50

Figure 1: The illustration of the CORL library design. Single-file implementation takes a yaml configuration file with both environment and algorithm parameters to run the experiment, which logs all required statistics to Weights&Biases (Biewald, 2020).

## 2  Related Work

Since the Atari breakthrough (Mnih et al., 2015), numerous open-source RL frameworks and libraries have been developed over the last years: (Dhariwal et al., 2017; Hill et al., 2018; Castro et al., 2018; Gauci et al., 2018; Keng & Graesser, 2017; garage contributors, 2019; Duan et al., 2016; Kolesnikov & Hrinchuk, 2019; Fujita et al., 2021; Liang et al., 2018; Fujita et al., 2021; Liu et al., 2021; Huang et al., 2021; Weng et al., 2021; Stooke & Abbeel, 2019), focusing on different perspectives of the RL. For example, stable-baselines (Hill et al., 2018) provides many deep RL implementations that carefully reproduce results to back up RL practitioners with reliable baselines during methods comparison. On the other hand, Ray (Liang et al., 2018) focuses on implementations scalability and production-friendly usage. Finally, more nuanced solutions exist, such as Dopamine (Castro et al., 2018), which emphasizes different DQN variants, or ReAgent (Gauci et al., 2018), which applies RL to the RecSys domain.

At the same time, the offline RL branch and especially offline-to-online, which we are interested in this paper, are not yet covered as much: the only library that precisely focuses on offline RL setting is d3rlpy (Takuma Seno, 2021). While CORL also focuses on offline RL methods (Nair et al., 2020; Kumar et al., 2020; Kostrikov et al., 2021; Fujimoto & Gu, 2021; An et al., 2021; Chen et al., 2021), similar to d3rlpy, it takes a different perspective on library design and provides *non-modular* independent algorithms implementations. More precisely, CORL does not introduce additional abstractions to make offline RL more general but instead gives an "easy-to-hack" starter kit for research needs. Finally, CORL also provides recent offline-to-online solutions (Nair et al., 2020; Kumar et al., 2020; Kostrikov et al., 2021; Wu et al., 2022; Nakamoto et al., 2023; Tarasov et al., 2023) that are gaining interest among researchers and practitioners.

Although CORL does not represent the first non-modular RL library, which is more likely the CleanRL (Huang et al., 2021) case, it has two significant differences from its predecessor. First, CORL is focused on *offline* and *offline-to-online* RL, while CleanRL implements *online* RL algorithms. Second, CORL intends to minimize the complexity of the requirements and external dependencies. To be more concrete, CORL does not have additional requirements with abstractions such as $stable\text{-}baselines$ (Hill et al., 2018) or $envpool$ (Weng et al., 2022) but instead implements everything from scratch in the codebase.

## 3 CORL Design

**Single-File Implementations**

Implementational subtleties significantly impact agent performance in deep RL (Henderson et al., 2018; Engstrom et al., 2020; Fujimoto & Gu, 2021). Unfortunately, user-friendly abstractions and general interfaces, the core idea behind modular libraries, encapsulate and often hide these important nuances from the practitioners. For such a reason, CORL unwraps these details by adopting single-file implementations. To be more concrete, we put environment details, algorithms hyperparameters, and evaluation parameters into a single file[4]. For example, we provide

- $any\_percent\_bc.py$ (404 LOC[5]) as a baseline algorithm for offline RL methods comparison,

- $td3\_bc.py$ (511 LOC) as a competitive minimalistic offline RL algorithm (Fujimoto & Gu, 2021),

- $dt.py$ (540 LOC) as an example of the recently proposed trajectory optimization approach (Chen et al., 2021)

Figure 1 depicts an overall library design. To avoid over-complicated offline implementations, we treat offline and offline-to-online versions of the same algorithms separately. While such design produces code duplications among realization, it has several essential benefits from the both educational and research perspective:

- **Smooth learning curve**. Having the entire code in one place makes understanding all its aspects more straightforward. In other words, one may find it easier to dive into 540 LOC of single-file Decision Transformer (Chen et al., 2021) implementation rather than 10+ files of the original implementation[6].

- **Simple prototyping**. As we are not interested in the code's general applicability, we could make it implementation-specific. Such a design also removes the need for inheritance from general primitives or their refactoring, reducing abstraction overhead to zero. At the same time, this idea gives us complete freedom during code modification.

- **Faster debugging**. Without additional abstractions, implementation simplifies to a single for-loop with a global Python name scope. Furthermore, such flat architecture makes accessing and inspecting any created variable easier during training, which is crucial in the presence of modifications and debugging.

**Configuration files**

Although it is a typical pattern to use a command line interface (CLI) for single-file experiments in the research community, CORL slightly improves it with predefined configuration files. Utilizing YAML parsing through CLI, for each experiment, we gather all environment and algorithm hyperparameters into such files so that one can use them as an initial setup. We found that such setup (1) simplifies experiments, eliminating the need to keep all algorithm-environment-specific parameters in mind, and (2) keeps it convenient with the familiar CLI approach.

---

[4]We follow the PEP8 style guide with a maximum line length of 89, which increases LOC a bit.

[5]Lines Of Code

[6]Original Decision Transformer implementation: `https://github.com/kzl/decision-transformer`

**Experiment Tracking**

Offline RL evaluation is another challenging aspect of the current offline RL state (Kurenkov & Kolesnikov, 2022). To face this uncertainty, CORL supports integration with Weights&Biases (Biewald, 2020), a modern experiment tracking tool. With each experiment, CORL automatically saves (1) source code, (2) dependencies (requirements.txt), (3) hardware setup, (4) OS environment variables, (5) hyperparameters, (6) training, and system metrics, (7) logs (stdout, stderr). See Appendix B for an example.

Although, Weights&Biases is a proprietary solution, other alternatives, such as Tensorboard (Abadi et al., 2015) or Aim (Arakelyan et al., 2020), could be used within a few lines of code change. It is also important to note that with Weights&Biases tracking, one could easily use CORL with sweeps or public reports.

We found full metrics tracking during the training process necessary for two reasons. First, it removes the possible bias of the final or best performance commonly reported in papers. For example, one could evaluate offline RL performance as max archived score, while another uses the average scores over $N$ (last) evaluations (Takuma Seno, 2021). Second, it provides an opportunity for advanced performance analysis such as EOP (Kurenkov & Kolesnikov, 2022) or RLiable (Agarwal et al., 2021). In short, when provided with all metrics logs, one can utilize all performance statistics, not merely relying on commonly used alternatives.

# 4 Benchmarking D4RL

## 4.1 Offline

In our library, we implemented the following offline algorithms: $N\%$[7] Behavioral Cloning (BC), TD3 + BC (Fujimoto & Gu, 2021), CQL (Kumar et al., 2020), IQL (Kostrikov et al., 2021), AWAC (Nair et al., 2020), ReBRAC (Tarasov et al., 2023), SAC-N, EDAC (An et al., 2021), and Decision Transformer (DT) (Chen et al., 2021). We evaluated every algorithm on the D4RL benchmark (Fu et al., 2020), focusing on Gym-MuJoCo, Maze2D, AntMaze, and Adroit tasks. Each algorithm was run for one million gradient steps[8] and evaluated using ten episodes for Gym-MuJoCo and Adroit tasks. For Maze2d, we use 100 evaluation episodes. In our experiments, we tried to rely on the hyperparameters proposed in the original works (see Appendix D for details) as much as possible.

The final performance is reported in Table 1 and the maximal performance in Table 2. The scores are normalized to the range between 0 and 100 (Fu et al., 2020). Following the recent work by Takuma Seno (2021), we report the last and best-obtained scores to illustrate each algorithm's potential performance and overfitting properties. Figure 2 shows the performance profiles and probability of improvement of ReBRAC over other algorithms (Agarwal et al., 2021). See Appendix A for complete training performance graphs.

Based on these results, we make several valuable observations. First, ReBRAC, IQL and AWAC are the most competitive baselines in offline setup on average. Note that AWAC is often omitted in recent works.

> **Observation 1**: ReBRAC, IQL and AWAC are the strongest offline baselines on average.

Second, EDAC outperforms all other algorithms on Gym-MuJoCo by a significant margin, and to our prior knowledge, there are still no algorithms that perform much better on these tasks. SAC-N shows the best performance on Maze2d tasks. However, simultaneously, SAC-N and EDAC cannot solve AntMaze tasks and perform poorly in the Adroit domain.

> **Observation 2**: SAC-N and EDAC are the strongest baselines for Gym-MuJoCo and Maze2d, but they perform poorly on both AntMaze and Adroit domains.

Third, during our experiments, we observed that the hyperparameters proposed for CQL in Kumar et al. (2020) do not perform as well as claimed on most tasks. CQL is extremely sensitive to the

---

[7]$N$ is a percentage of best trajectories with the highest return used for training. We omit the percentage when it is equal to 100.

[8]Except SAC-$N$, EDAC, and DT due to their original hyperparameters. See Appendix D for details.

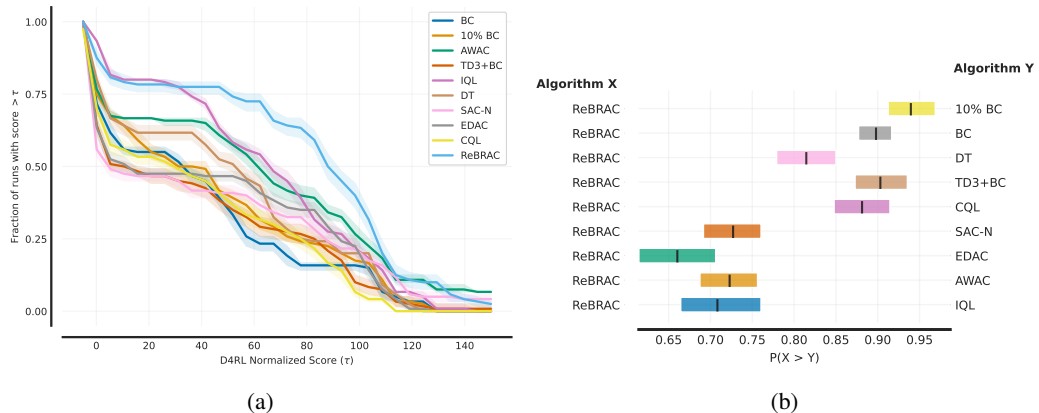

Figure 2: (a) Performance profiles after offline training (b) Probability of improvement of ReBRAC to other algorithms after offline training. The curves (Agarwal et al., 2021) are for D4RL benchmark spanning Gym-MuJoCo, Maze2d, AntMaze, and Adroit datasets.

Table 1: Normalized performance of the last trained policy on D4RL averaged over 4 random seeds.

| Task Name | BC | BC-10% | TD3+BC | AWAC | CQL | IQL | ReBRAC | SAC-N | EDAC | DT |
|---|---|---|---|---|---|---|---|---|---|---|
| halfcheetah-medium-v2 | 42.40 ± 0.19 | 42.46 ± 0.70 | 48.10 ± 0.18 | 50.02 ± 0.27 | 47.04 ± 0.22 | 48.31 ± 0.22 | 64.04 ± 0.68 | **68.20** ± 1.28 | 67.70 ± 1.04 | 42.20 ± 0.26 |
| halfcheetah-medium-replay-v2 | 35.66 ± 2.33 | 23.59 ± 6.95 | 44.84 ± 0.59 | 45.13 ± 0.88 | 45.04 ± 0.27 | 44.46 ± 0.22 | 51.18 ± 0.31 | 60.70 ± 1.01 | **62.06** ± 1.10 | 38.91 ± 0.50 |
| halfcheetah-medium-expert-v2 | 55.95 ± 7.35 | 90.10 ± 2.45 | 90.78 ± 6.04 | 95.00 ± 0.61 | 95.63 ± 0.42 | 94.74 ± 0.52 | 103.80 ± 2.95 | 98.96 ± 9.31 | **104.76** ± 0.64 | 91.55 ± 0.95 |
| hopper-medium-v2 | 53.51 ± 1.76 | 55.48 ± 7.30 | 60.37 ± 3.49 | 63.02 ± 4.56 | 59.08 ± 3.77 | 67.53 ± 3.78 | **102.29** ± 0.17 | 40.82 ± 9.91 | 101.70 ± 0.28 | 65.10 ± 1.61 |
| hopper-medium-replay-v2 | 29.81 ± 2.07 | 70.42 ± 8.66 | 64.42 ± 21.52 | 98.88 ± 2.07 | 95.11 ± 5.27 | 97.43 ± 6.39 | 94.98 ± 6.53 | **100.33** ± 0.78 | 99.66 ± 0.81 | 81.77 ± 6.87 |
| hopper-medium-expert-v2 | 52.30 ± 4.01 | 111.16 ± 1.03 | 101.17 ± 9.07 | 101.90 ± 6.22 | 99.26 ± 10.91 | 107.42 ± 7.80 | 109.45 ± 2.34 | 101.31 ± 11.63 | 105.19 ± 10.08 | **110.44** ± 0.33 |
| walker2d-medium-v2 | 63.23 ± 16.24 | 67.34 ± 5.17 | 82.71 ± 4.78 | 68.52 ± 27.19 | 80.75 ± 3.28 | 80.91 ± 3.17 | 85.82 ± 0.77 | 87.47 ± 0.66 | **93.36** ± 1.38 | 67.63 ± 2.54 |
| walker2d-medium-replay-v2 | 21.80 ± 10.15 | 54.35 ± 6.34 | 85.62 ± 4.01 | 80.62 ± 3.58 | 73.09 ± 13.22 | 82.15 ± 3.03 | 84.25 ± 2.25 | 78.99 ± 0.50 | **87.10** ± 2.78 | 59.86 ± 2.73 |
| walker2d-medium-expert-v2 | 98.96 ± 15.98 | 108.70 ± 0.25 | 110.03 ± 0.36 | 111.44 ± 1.62 | 109.56 ± 0.39 | 111.72 ± 0.86 | 111.86 ± 0.43 | **114.93** ± 0.41 | 114.75 ± 0.74 | 107.11 ± 0.96 |
| **Gym-MuJoCo avg** | 50.40 | 69.29 | 76.45 | 79.39 | 78.28 | 81.63 | 89.74 | 83.52 | **92.92** | 73.84 |
| maze2d-umaze-v1 | 0.36 ± 8.69 | 12.18 ± 4.29 | 29.41 ± 12.31 | 65.65 ± 5.34 | -8.90 ± 6.11 | 42.11 ± 0.58 | 106.87 ± 22.16 | **130.59** ± 16.52 | 95.26 ± 6.39 | 18.08 ± 25.42 |
| maze2d-medium-v1 | 0.79 ± 3.25 | 14.25 ± 2.33 | 59.45 ± 36.25 | 84.63 ± 35.54 | 86.11 ± 9.68 | 34.85 ± 2.72 | **105.11** ± 31.67 | 88.61 ± 18.72 | 57.04 ± 3.45 | 31.71 ± 26.33 |
| maze2d-large-v1 | 2.26 ± 4.39 | 11.32 ± 5.10 | 97.10 ± 25.41 | **215.50** ± 3.11 | 23.75 ± 36.70 | 61.72 ± 3.50 | 78.33 ± 61.77 | 204.76 ± 1.19 | 95.60 ± 22.92 | 35.66 ± 28.20 |
| **Maze2d avg** | 1.13 | 12.58 | 61.99 | 121.92 | 33.65 | 46.23 | 96.77 | **141.32** | 82.64 | 28.48 |
| antmaze-umaze-v2 | 55.25 ± 4.15 | 65.75 ± 5.26 | 70.75 ± 39.18 | 56.75 ± 9.09 | 92.75 ± 1.92 | 77.00 ± 5.52 | **97.75** ± 1.48 | 0.00 ± 0.00 | 0.00 ± 0.00 | 57.00 ± 9.82 |
| antmaze-umaze-diverse-v2 | 47.25 ± 4.09 | 44.00 ± 1.00 | 44.75 ± 11.61 | 54.75 ± 8.01 | 37.25 ± 3.70 | 54.25 ± 5.54 | **83.50** ± 7.02 | 0.00 ± 0.00 | 0.00 ± 0.00 | 51.75 ± 0.43 |
| antmaze-medium-play-v2 | 0.00 ± 0.00 | 2.00 ± 0.71 | 0.25 ± 0.43 | 0.00 ± 0.00 | 65.75 ± 11.61 | 65.75 ± 11.71 | **89.50** ± 3.35 | 0.00 ± 0.00 | 0.00 ± 0.00 | 0.00 ± 0.00 |
| antmaze-medium-diverse-v2 | 0.75 ± 0.83 | 5.75 ± 9.39 | 0.25 ± 0.43 | 0.00 ± 0.00 | 67.25 ± 3.56 | 73.75 ± 5.45 | **83.50** ± 8.20 | 0.00 ± 0.00 | 0.00 ± 0.00 | 0.00 ± 0.00 |
| antmaze-large-play-v2 | 0.00 ± 0.00 | 0.00 ± 0.00 | 0.00 ± 0.00 | 0.00 ± 0.00 | 20.75 ± 7.26 | 42.00 ± 4.53 | **52.25** ± 29.01 | 0.00 ± 0.00 | 0.00 ± 0.00 | 0.00 ± 0.00 |
| antmaze-large-diverse-v2 | 0.00 ± 0.00 | 0.75 ± 0.83 | 0.00 ± 0.00 | 0.00 ± 0.00 | 20.50 ± 13.24 | 30.25 ± 3.63 | **64.00** ± 5.43 | 0.00 ± 0.00 | 0.00 ± 0.00 | 0.00 ± 0.00 |
| **AntMaze avg** | 17.21 | 19.71 | 19.33 | 18.58 | 50.71 | 57.17 | **78.42** | 0.00 | 0.00 | 18.12 |
| pen-human-v1 | 71.03 ± 6.26 | 26.99 ± 9.60 | -3.88 ± 0.21 | 76.65 ± 11.71 | 13.71 ± 16.98 | 78.49 ± 8.21 | **103.16** ± 8.49 | 6.86 ± 5.93 | 5.07 ± 6.16 | 67.68 ± 5.48 |
| pen-cloned-v1 | 51.92 ± 15.15 | 46.67 ± 14.25 | 5.13 ± 5.28 | 85.72 ± 16.92 | 1.04 ± 6.62 | 83.42 ± 8.19 | **102.79** ± 7.84 | 31.35 ± 2.14 | 12.02 ± 1.75 | 64.43 ± 1.43 |
| pen-expert-v1 | 109.65 ± 7.28 | 114.96 ± 2.96 | 122.53 ± 21.27 | **159.91** ± 1.87 | -1.41 ± 2.34 | 128.05 ± 9.21 | 152.16 ± 6.33 | 87.11 ± 48.95 | -1.55 ± 0.81 | 116.38 ± 1.27 |
| door-human-v1 | 2.34 ± 4.00 | -0.13 ± 0.07 | -0.33 ± 0.01 | 2.39 ± 2.26 | **5.53** ± 1.31 | 3.26 ± 1.83 | -0.10 ± 0.01 | -0.38 ± 0.00 | -0.12 ± 0.13 | 4.44 ± 0.87 |
| door-cloned-v1 | -0.09 ± 0.03 | 0.29 ± 0.59 | -0.34 ± 0.01 | -0.01 ± 0.01 | -0.33 ± 0.01 | 3.07 ± 1.75 | 0.06 ± 0.05 | -0.33 ± 0.00 | 2.66 ± 2.31 | **7.64** ± 3.26 |
| door-expert-v1 | 105.35 ± 0.09 | 104.04 ± 1.46 | -0.33 ± 0.01 | 104.57 ± 0.31 | -0.32 ± 0.02 | **106.65** ± 0.25 | 106.37 ± 0.29 | -0.33 ± 0.00 | 106.29 ± 1.73 | 104.87 ± 0.39 |
| hammer-human-v1 | 3.03 ± 3.39 | -0.19 ± 0.02 | 1.02 ± 0.24 | 1.01 ± 0.51 | 0.14 ± 0.11 | **1.79** ± 0.80 | 0.24 ± 0.24 | 0.24 ± 0.00 | 0.28 ± 0.18 | 1.28 ± 0.15 |
| hammer-cloned-v1 | 0.55 ± 0.16 | 0.12 ± 0.08 | 0.25 ± 0.01 | 1.27 ± 2.11 | 0.30 ± 0.01 | 1.50 ± 0.69 | **5.00** ± 3.75 | 0.14 ± 0.09 | 0.19 ± 0.07 | 1.82 ± 0.55 |
| hammer-expert-v1 | 126.78 ± 0.64 | 121.75 ± 7.67 | 3.11 ± 0.03 | 127.08 ± 0.13 | 0.26 ± 0.01 | 128.68 ± 0.33 | **133.62** ± 0.27 | 25.13 ± 43.25 | 28.52 ± 49.00 | 117.45 ± 6.65 |
| relocate-human-v1 | 0.04 ± 0.03 | -0.14 ± 0.08 | -0.29 ± 0.01 | 0.45 ± 0.53 | 0.06 ± 0.03 | 0.12 ± 0.04 | **0.16** ± 0.30 | -0.31 ± 0.01 | -0.17 ± 0.17 | 0.05 ± 0.01 |
| relocate-cloned-v1 | -0.06 ± 0.01 | -0.00 ± 0.02 | -0.30 ± 0.01 | -0.01 ± 0.03 | -0.29 ± 0.01 | 0.04 ± 0.01 | **1.66** ± 2.59 | -0.01 ± 0.10 | 0.17 ± 0.35 | 0.16 ± 0.09 |
| relocate-expert-v1 | 107.58 ± 1.20 | 97.90 ± 5.21 | -1.73 ± 0.96 | **109.52** ± 0.47 | -0.30 ± 0.02 | 106.11 ± 4.02 | 107.52 ± 2.28 | -0.36 ± 0.00 | 71.94 ± 18.37 | 104.28 ± 0.42 |
| **Adroit avg** | 48.18 | 42.69 | 10.40 | 55.71 | 1.53 | 53.43 | **59.39** | 12.43 | 18.78 | 49.21 |
| **Total avg** | 37.95 | 43.06 | 37.16 | 62.01 | 37.61 | 61.92 | **76.04** | 44.16 | 43.65 | 48.31 |

choice of hyperparameters, and we had to tune them a lot to make it work on each domain (see Table 7). For example, AntMaze requires five hidden layers for the critic networks, while other tasks' performance suffers with this number of layers. The issue of sensitivity[9] was already mentioned in prior works as well (An et al., 2021; Ghasemipour et al., 2022).

> **Observation 3**: CQL is extremely sensitive to the choice of hyperparameters and implementation details.

Fourth, we also observe that the hyperparameters do not always work the same way when transferring between Deep Learning frameworks [10]. Our implementations of IQL and CQL use PyTorch, but the parameters from reference JAX implementations sometimes strongly underperform (e.g., IQL on Hopper tasks and CQL on Adroit).

---

[9]See also `https://github.com/aviralkumar2907/CQL/issues/9`, `https://github.com/tinkoff-ai/CORL/issues/14` and `https://github.com/young-geng/CQL/issues/5`

[10]`https://github.com/tinkoff-ai/CORL/issues/33`

Table 2: Normalized performance of the best trained policy on D4RL averaged over 4 random seeds.

| Task Name | BC | BC-10% | TD3+BC | AWAC | CQL | IQL | ReBRAC | SAC-$N$ | EDAC | DT |
|---|---|---|---|---|---|---|---|---|---|---|
| halfcheetah-medium-v2 | 43.60 ± 0.14 | 43.90 ± 0.13 | 48.93 ± 0.11 | 50.81 ± 0.15 | 47.62 ± 0.03 | 48.84 ± 0.07 | 65.62 ± 0.46 | **72.21** ± 0.31 | 69.72 ± 0.92 | 42.73 ± 0.10 |
| halfcheetah-medium-replay-v2 | 40.52 ± 0.19 | 42.27 ± 0.46 | 45.84 ± 0.26 | 46.47 ± 0.26 | 46.43 ± 0.19 | 45.35 ± 0.08 | 52.22 ± 0.31 | **67.29** ± 0.34 | 66.55 ± 1.05 | 40.31 ± 0.28 |
| halfcheetah-medium-expert-v2 | 79.69 ± 3.10 | 94.11 ± 0.22 | 96.59 ± 0.87 | 96.83 ± 0.23 | 97.04 ± 0.17 | 95.38 ± 0.17 | 108.89 ± 1.20 | **111.73** ± 0.47 | 110.62 ± 1.04 | 93.40 ± 0.21 |
| hopper-medium-v2 | 69.04 ± 2.90 | 73.84 ± 0.37 | 70.44 ± 1.18 | 95.42 ± 3.67 | 70.80 ± 1.98 | 80.46 ± 3.09 | 103.19 ± 0.16 | 101.79 ± 0.20 | **103.26** ± 0.14 | 69.42 ± 3.64 |
| hopper-medium-replay-v2 | 68.88 ± 10.33 | 90.57 ± 2.07 | 98.12 ± 1.16 | 101.47 ± 0.23 | 101.63 ± 0.55 | 102.69 ± 0.96 | 102.57 ± 0.45 | **103.83** ± 0.53 | 103.28 ± 0.49 | 88.74 ± 3.02 |
| hopper-medium-expert-v2 | 90.63 ± 10.98 | 113.13 ± 0.16 | 113.22 ± 0.43 | **113.26** ± 0.49 | 112.84 ± 0.66 | 113.18 ± 0.38 | 113.16 ± 0.43 | 111.24 ± 0.15 | 111.80 ± 0.11 | 111.18 ± 0.21 |
| walker2d-medium-v2 | 80.64 ± 0.91 | 82.05 ± 0.93 | 86.91 ± 0.28 | 85.86 ± 3.76 | 84.77 ± 0.20 | 87.58 ± 0.48 | 87.79 ± 0.19 | 90.17 ± 0.54 | **95.78** ± 1.07 | 74.70 ± 0.56 |
| walker2d-medium-replay-v2 | 48.41 ± 7.61 | 76.09 ± 0.40 | **91.17** ± 0.72 | 86.70 ± 0.94 | 89.39 ± 0.88 | 89.94 ± 0.93 | 91.11 ± 0.63 | 85.18 ± 1.63 | 89.69 ± 1.39 | 68.22 ± 1.20 |
| walker2d-medium-expert-v2 | 109.95 ± 0.62 | 109.90 ± 0.09 | 112.21 ± 0.06 | 113.40 ± 2.22 | 111.63 ± 0.38 | 113.06 ± 0.53 | 112.49 ± 0.18 | **116.93** ± 0.42 | 116.52 ± 0.75 | 108.71 ± 0.34 |
| **Gym-MuJoCo avg** | 70.15 | 80.65 | 84.83 | 87.80 | 84.68 | 86.28 | 93.00 | 95.60 | **96.36** | 77.49 |
| maze2d-umaze-v1 | 16.09 ± 0.87 | 22.49 ± 1.52 | 99.33 ± 16.16 | 136.96 ± 10.89 | 92.05 ± 13.66 | 50.92 ± 4.23 | **162.28** ± 1.79 | 153.12 ± 6.49 | 149.88 ± 1.97 | 63.83 ± 17.35 |
| maze2d-medium-v1 | 19.16 ± 1.24 | 27.64 ± 1.87 | 150.93 ± 3.89 | 152.73 ± 20.78 | 128.66 ± 5.44 | 122.69 ± 30.00 | 150.12 ± 4.48 | 93.80 ± 14.66 | **154.41** ± 1.58 | 68.14 ± 12.25 |
| maze2d-large-v1 | 20.75 ± 6.66 | 41.83 ± 3.64 | 197.64 ± 5.26 | **227.31** ± 1.47 | 157.51 ± 7.32 | 162.25 ± 44.18 | 197.55 ± 5.82 | 207.51 ± 0.96 | 182.52 ± 2.68 | 50.25 ± 19.34 |
| **Maze2d avg** | 18.67 | 30.65 | 149.30 | 172.33 | 126.07 | 111.95 | 169.98 | 151.48 | 162.27 | 60.74 |
| antmaze-umaze-v2 | 68.50 ± 2.29 | 77.50 ± 1.50 | 98.50 ± 0.87 | 70.75 ± 8.84 | 94.75 ± 0.83 | 84.00 ± 4.06 | **100.00** ± 0.00 | 0.00 ± 0.00 | 42.50 ± 28.61 | 64.50 ± 2.06 |
| antmaze-umaze-diverse-v2 | 64.75 ± 4.32 | 63.50 ± 2.18 | 71.25 ± 5.76 | 81.50 ± 4.27 | 53.75 ± 2.05 | 79.50 ± 3.35 | **96.75** ± 2.28 | 0.00 ± 0.00 | 0.00 ± 0.00 | 60.50 ± 2.29 |
| antmaze-medium-play-v2 | 4.50 ± 1.12 | 6.25 ± 2.38 | 3.75 ± 1.30 | 25.00 ± 10.70 | 80.50 ± 3.35 | 78.50 ± 3.84 | **93.50** ± 2.60 | 0.00 ± 0.00 | 0.00 ± 0.00 | 0.75 ± 0.43 |
| antmaze-medium-diverse-v2 | 4.75 ± 1.09 | 16.50 ± 5.59 | 5.50 ± 1.50 | 10.75 ± 5.31 | 71.00 ± 4.53 | 83.50 ± 1.80 | **91.75** ± 2.05 | 0.00 ± 0.00 | 0.00 ± 0.00 | 0.50 ± 0.50 |
| antmaze-large-play-v2 | 0.50 ± 0.50 | 13.50 ± 9.76 | 1.25 ± 0.43 | 0.50 ± 0.50 | 34.75 ± 5.85 | 53.50 ± 2.50 | **68.75** ± 13.90 | 0.00 ± 0.00 | 0.00 ± 0.00 | 0.00 ± 0.00 |
| antmaze-large-diverse-v2 | 0.75 ± 0.43 | 6.25 ± 1.79 | 0.25 ± 0.43 | 0.00 ± 0.00 | 36.25 ± 3.34 | 53.00 ± 3.00 | **69.50** ± 7.26 | 0.00 ± 0.00 | 0.00 ± 0.00 | 0.00 ± 0.00 |
| **AntMaze avg** | 23.96 | 30.58 | 30.08 | 31.42 | 61.83 | 72.00 | **86.71** | 0.00 | 7.08 | 21.04 |
| pen-human-v1 | 99.69 ± 7.45 | 59.89 ± 8.03 | 9.95 ± 8.19 | 119.03 ± 6.55 | 58.91 ± 1.81 | 106.15 ± 10.28 | **127.28** ± 3.22 | 56.48 ± 7.17 | 35.84 ± 10.57 | 77.83 ± 2.30 |
| pen-cloned-v1 | 99.14 ± 12.27 | 83.62 ± 11.75 | 52.66 ± 6.33 | 125.78 ± 3.28 | 14.74 ± 2.31 | 114.05 ± 4.78 | **128.64** ± 7.15 | 52.69 ± 5.30 | 26.90 ± 7.85 | 71.17 ± 2.70 |
| pen-expert-v1 | 128.77 ± 5.88 | 134.36 ± 3.16 | 142.83 ± 7.72 | **162.53** ± 0.30 | 14.86 ± 4.07 | 140.01 ± 6.36 | 157.62 ± 0.26 | 116.43 ± 40.26 | 36.04 ± 4.60 | 119.49 ± 2.31 |
| door-human-v1 | 9.41 ± 4.55 | 7.00 ± 6.77 | -0.11 ± 0.06 | **17.70** ± 2.55 | 13.28 ± 2.77 | 13.52 ± 1.22 | 0.27 ± 0.43 | -0.10 ± 0.06 | 2.51 ± 2.26 | 7.36 ± 1.24 |
| door-cloned-v1 | 3.40 ± 0.95 | 10.37 ± 4.09 | -0.20 ± 0.11 | 10.53 ± 2.82 | -0.08 ± 0.13 | 9.02 ± 1.47 | 7.73 ± 6.80 | -0.21 ± 0.10 | **20.36** ± 1.11 | 11.18 ± 0.96 |
| door-expert-v1 | 105.84 ± 0.23 | 105.92 ± 0.24 | 4.49 ± 7.39 | 106.60 ± 0.27 | 59.47 ± 25.04 | 107.29 ± 0.37 | 106.78 ± 0.04 | 0.05 ± 0.02 | **109.22** ± 0.24 | 105.49 ± 0.09 |
| hammer-human-v1 | 12.61 ± 4.87 | 6.23 ± 4.79 | 2.38 ± 0.14 | **16.95** ± 3.61 | 0.30 ± 0.05 | 6.86 ± 2.38 | 1.18 ± 0.15 | 0.25 ± 0.00 | 3.49 ± 2.17 | 1.68 ± 0.11 |
| hammer-cloned-v1 | 8.90 ± 4.04 | 8.72 ± 3.28 | 0.96 ± 0.30 | 10.74 ± 5.54 | 0.32 ± 0.03 | 11.63 ± 1.70 | **48.16** ± 6.20 | 12.67 ± 15.02 | 0.27 ± 0.01 | 2.74 ± 0.22 |
| hammer-expert-v1 | 127.89 ± 0.57 | 128.15 ± 0.66 | 33.31 ± 47.65 | 129.08 ± 0.26 | 0.93 ± 1.12 | 129.76 ± 0.37 | **134.74** ± 0.30 | 91.74 ± 47.77 | 69.44 ± 47.00 | 127.39 ± 0.10 |
| relocate-human-v1 | 0.59 ± 0.27 | 0.16 ± 0.14 | -0.29 ± 0.01 | 1.77 ± 0.84 | 1.03 ± 0.20 | 1.22 ± 0.28 | **3.70** ± 2.34 | -0.18 ± 0.14 | 0.05 ± 0.02 | 0.08 ± 0.02 |
| relocate-cloned-v1 | 0.45 ± 0.31 | 0.74 ± 0.45 | -0.02 ± 0.04 | 0.39 ± 0.13 | -0.07 ± 0.02 | 1.78 ± 0.70 | **9.25** ± 2.56 | 0.10 ± 0.04 | 4.11 ± 1.39 | 0.34 ± 0.09 |
| relocate-expert-v1 | 110.31 ± 0.36 | 109.77 ± 0.60 | 0.23 ± 0.27 | **111.21** ± 0.32 | 0.03 ± 0.10 | 110.12 ± 0.82 | 111.14 ± 0.23 | -0.07 ± 0.08 | 98.32 ± 3.75 | 106.49 ± 0.30 |
| **Adroit avg** | 58.92 | 54.58 | 20.51 | 67.69 | 13.65 | 62.62 | **69.71** | 27.49 | 33.88 | 52.60 |
| **Total avg** | 51.27 | 55.21 | 54.60 | 76.93 | 55.84 | 76.53 | **90.12** | 54.82 | 60.10 | 54.57 |

> **Observation 4**: Hyperparameters are not always transferable between Deep Learning frameworks.

## 4.2 Offline-to-Online

We also implement the following algorithms in offline-to-online setup: CQL (Kumar et al., 2020), IQL (Kostrikov et al., 2021), AWAC (Nair et al., 2020), SPOT (Wu et al., 2022) Cal-QL (Nakamoto et al., 2023), ReBRAC (Tarasov et al., 2023). Inspired by Nakamoto et al. (2023), we evaluate algorithms on AntMaze and Adroit Cloned datasets[11]. Each algorithm is trained offline over 1 million steps and tuned using online transitions over another 1 million steps. The AntMaze tasks are evaluated using 100 episodes, while the Adroit tasks are tested with ten episodes.

Table 3: Normalized performance of algorithms after offline pretraining and online finetuning on D4RL averaged over 4 random seeds.

| Task Name | AWAC | CQL | IQL | SPOT | Cal-QL | ReBRAC |
|---|---|---|---|---|---|---|
| antmaze-umaze-v2 | 52.75 ± 8.67 → 98.75 ± 1.09 | 94.00 ± 1.58 → 99.50 ± 0.87 | 77.00 ± 0.71 → 96.50 ± 1.12 | 91.00 ± 2.55 → 99.50 ± 0.50 | 76.75 ± 7.53 → **99.75** ± 0.43 | 98.00 ± 1.82 → 74.75 ± 49.17 |
| antmaze-umaze-diverse-v2 | 56.00 ± 2.74 → 0.00 ± 0.00 | 9.50 ± 9.91 → **99.00** ± 1.22 | 59.50 ± 9.55 → 63.75 ± 25.02 | 36.25 ± 2.17 → 95.00 ± 3.67 | 32.00 ± 27.79 → 98.50 ± 1.12 | 73.75 ± 15.32 → 98.0 ± 3.36 |
| antmaze-medium-play-v2 | 0.00 ± 0.00 → 0.00 ± 0.00 | 59.00 ± 11.18 → 97.75 ± 1.30 | 71.75 ± 2.95 → 89.75 ± 1.09 | 67.25 ± 10.47 → 97.25 ± 1.30 | 71.75 ± 3.27 → **98.75** ± 1.64 | 87.5 ± 4.35 → 98.0 ± 1.82 |
| antmaze-medium-diverse-v2 | 0.00 ± 0.00 → 0.00 ± 0.00 | 63.50 ± 6.84 → 97.25 ± 1.92 | 64.25 ± 1.92 → 92.25 ± 2.86 | 73.75 ± 7.29 → 94.50 ± 1.66 | 62.00 ± 4.30 → **98.25** ± 1.48 | 85.25 ± 2.5 → 98.75 ± 0.5 |
| antmaze-large-play-v2 | 0.00 ± 0.00 → 0.00 ± 0.00 | 28.75 ± 7.76 → 88.25 ± 2.28 | 38.50 ± 8.73 → 64.50 ± 17.04 | 31.50 ± 12.58 → 87.00 ± 3.24 | 31.75 ± 8.87 → **97.25** ± 1.79 | 68.5 ± 7.1 → 31.5 ± 38.75 |
| antmaze-large-diverse-v2 | 0.00 ± 0.00 → 0.00 ± 0.00 | 35.50 ± 3.64 → **91.75** ± 3.96 | 26.75 ± 3.77 → 64.25 ± 4.15 | 17.50 ± 7.26 → 81.00 ± 14.14 | 44.00 ± 8.69 → 91.50 ± 3.91 | 67.0 ± 12.24 → 72.25 ± 48.18 |
| **AntMaze avg** | 18.12 → 16.46 (-1.66) | 48.38 → 95.58 (+47.20) | 56.29 → 78.50 (+22.21) | 52.88 → 92.38 (+39.50) | 53.04 → **97.33** (+24.29) | 79.99 → 78.87(-1.11) |
| pen-cloned-v1 | 88.66 ± 15.10 → 86.82 ± 11.12 | -2.76 ± 0.08 → -1.28 ± 2.16 | 84.19 ± 3.96 → **102.02** ± 20.75 | 6.19 ± 5.21 → 43.63 ± 20.09 | -2.66 ± 0.04 → -2.68 ± 0.12 | 74.04 ± 13.82 → 138.15 ± 3.71 |
| door-cloned-v1 | 0.93 ± 1.66 → 0.01 ± 0.00 | -0.33 ± 0.01 → -0.33 ± 0.01 | 1.19 ± 0.93 → **20.34** ± 9.32 | -0.21 ± 0.14 → 0.02 ± 0.31 | -0.33 ± 0.01 → -0.33 ± 0.01 | 0.06 ± 0.04 → 102.38 ± 9.54 |
| hammer-cloned-v1 | 1.80 ± 3.01 → 0.24 ± 0.04 | 0.56 ± 0.55 → 2.85 ± 4.81 | 1.35 ± 0.32 → **57.27** ± 28.49 | 3.97 ± 6.39 → 3.73 ± 4.99 | 0.25 ± 0.04 → 0.17 ± 0.17 | 6.53 ± 3.86 → 124.65 ± 8.51 |
| relocate-cloned-v1 | -0.04 ± 0.04 → -0.04 ± 0.01 | -0.33 ± 0.01 → -0.33 ± 0.01 | 0.04 ± 0.04 → **0.32** ± 0.38 | -0.24 ± 0.01 → -0.15 ± 0.05 | -0.31 ± 0.05 → -0.31 ± 0.04 | 0.69 ± 0.71 → 6.96 ± 5.3 |
| **Adroit Avg** | 22.84 → 21.76 (-1.08) | -0.72 → 0.22 (+0.94) | 21.69 → **44.99** (+23.3) | 2.43 → 11.81 (+9.38) | -0.76 → -0.79 (-0.03) | 20.33 → 93.03 (+72.7) |
| **Total avg** | 20.01 → 18.58 (-1.43) | 28.74 → 57.44 (+28.7) | 42.45 → **65.10** (+22.65) | 32.70 → 60.15 (+27.45) | 31.52 → 58.08 (+26.56) | 56.12 → 84.53 (+28.41) |

The scores, normalized after the offline stage and online tuning, are reported in Table 3. We also provide finetuning cumulative regret proposed by Nakamoto et al. (2023) in Table 4. Cumulative regret is calculated as $(1 - average\ success\ rate)^{12}$. It is bounded between 0 and 1, indicating the range of possible values. Lower values of cumulative regret indicate better algorithm efficiency. The performance profiles and probability of improvement of ReBRAC over other algorithms after online finetuning are presented in Figure 3.

---

[11] Note, Nakamoto et al. (2023) used modified Cloned datasets while we employ original data from D4RL because these datasets are more common to for benchmarking.

[12] As specified by the authors: `https://github.com/nakamotoo/Cal-QL/issues/1`

Table 4: Cumulative regret of online finetuning calculated as $1 - average\ success\ rate$ averaged over 4 random seeds.

| Task Name | AWAC | CQL | IQL | SPOT | Cal-QL | ReBRAC |
|---|---|---|---|---|---|---|
| antmaze-umaze-v2 | $0.04 \pm 0.01$ | $0.02 \pm 0.00$ | $0.07 \pm 0.00$ | $0.02 \pm 0.00$ | $\mathbf{0.01} \pm 0.00$ | $0.10 \pm 0.20$ |
| antmaze-umaze-diverse-v2 | $0.88 \pm 0.01$ | $0.09 \pm 0.01$ | $0.43 \pm 0.11$ | $0.22 \pm 0.07$ | $\mathbf{0.05} \pm 0.01$ | $0.04 \pm 0.02$ |
| antmaze-medium-play-v2 | $1.00 \pm 0.00$ | $0.08 \pm 0.01$ | $0.09 \pm 0.01$ | $0.06 \pm 0.00$ | $\mathbf{0.04} \pm 0.01$ | $0.02 \pm 0.00$ |
| antmaze-medium-diverse-v2 | $1.00 \pm 0.00$ | $0.08 \pm 0.00$ | $0.10 \pm 0.01$ | $0.05 \pm 0.01$ | $\mathbf{0.04} \pm 0.01$ | $0.03 \pm 0.00$ |
| antmaze-large-play-v2 | $1.00 \pm 0.00$ | $0.21 \pm 0.02$ | $0.34 \pm 0.05$ | $0.29 \pm 0.07$ | $\mathbf{0.13} \pm 0.02$ | $0.14 \pm 0.05$ |
| antmaze-large-diverse-v2 | $1.00 \pm 0.00$ | $0.21 \pm 0.03$ | $0.41 \pm 0.03$ | $0.23 \pm 0.08$ | $\mathbf{0.13} \pm 0.02$ | $0.29 \pm 0.45$ |
| **AntMaze avg** | 0.82 | 0.11 | 0.24 | 0.15 | **0.07** | 0.10 |
| pen-cloned-v1 | $0.46 \pm 0.02$ | $0.97 \pm 0.00$ | $\mathbf{0.37} \pm 0.01$ | $0.58 \pm 0.02$ | $0.98 \pm 0.01$ | $0.08 \pm 0.01$ |
| door-cloned-v1 | $1.00 \pm 0.00$ | $1.00 \pm 0.00$ | $\mathbf{0.83} \pm 0.03$ | $0.99 \pm 0.01$ | $1.00 \pm 0.00$ | $0.18 \pm 0.06$ |
| hammer-cloned-v1 | $1.00 \pm 0.00$ | $1.00 \pm 0.00$ | $\mathbf{0.65} \pm 0.10$ | $0.98 \pm 0.01$ | $1.00 \pm 0.00$ | $0.12 \pm 0.03$ |
| relocate-cloned-v1 | $1.00 \pm 0.00$ | $1.00 \pm 0.00$ | $1.00 \pm 0.00$ | $1.00 \pm 0.00$ | $1.00 \pm 0.00$ | $0.9 \pm 0.06$ |
| **Adroit avg** | 0.86 | 0.99 | **0.71** | 0.89 | 0.99 | 0.32 |
| **Total avg** | 0.84 | 0.47 | **0.43** | 0.44 | 0.44 | 0.19 |

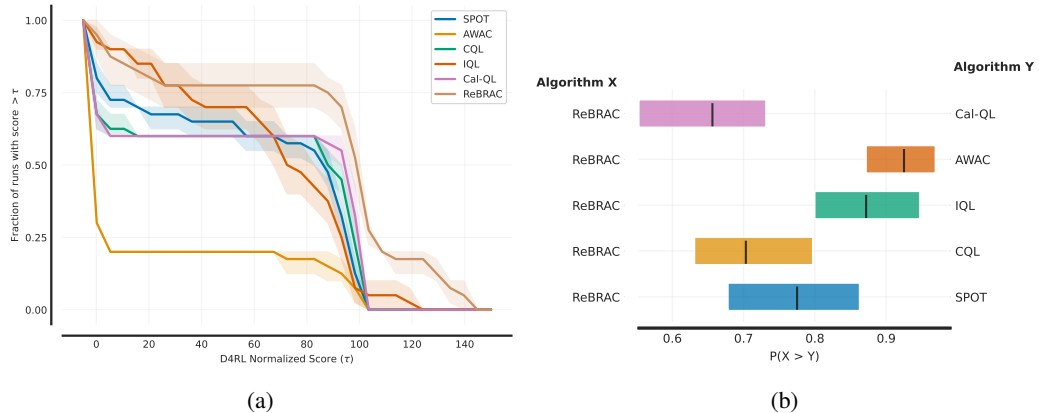

(a)          (b)

Figure 3: (a) Performance profiles after online tuning (b) Probability of improvement of ReBRAC to other algorithms after online tuning. The curves (Agarwal et al., 2021) are for D4RL benchmark spanning AntMaze and Adroit cloned datasets.

AWAC, initially proposed for finetuning purposes, appeared to be the worst of the considered algorithms, where the score is improved only on the most straightforward antmaze-umaze-v2 dataset. At the same time, on other datasets, performances either stay the same or even drop.

> **Observation 5**: AWAC does not benefit from online tuning on the considered tasks.

Cal-QL was proposed as a modification of CQL, which is expected to work better in offline-to-online setting. However, in our experiments, after finetuning CQL obtained scores which are not very different from Cal-QL. At the same time, we could not make both algorithms solve Adroit tasks[13].

> **Observation 6**: There is no big difference between CQL and Cal-QL. On AntMaze, these algorithms perform the best but work poorly on Adroit.

IQL starts with good offline scores on AntMaze, but it is less efficient in finetuning than other algorithms except for AWAC. At the same time, IQL and ReBRAC are the only algorithms that notably improve its scores after tuning on Adroit tasks, making them the most competitive offline-to-online baselines considering the average score.

---

[13]The issues are Observations 3 and 4. Additional hyperparameters search is needed.

> **Observation 7**: Considering offline and offline-to-online results, IQL and ReBRAC appear to be the strongest baselines on average.

## 5  Conclusion

This paper introduced CORL, a single-file implementation library for offline and offline-to-online reinforcement learning algorithms with configuration files and advanced metrics tracking support. In total, we provided implementations of ten offline and six offline-to-online algorithms. All implemented approaches were benchmarked on D4RL datasets, closely matching (sometimes overperforming) the reference results, if available. Focusing on implementation clarity and reproducibility, we hope that CORL will help RL practitioners in their research and applications.

This study's benchmarking results and observations are intended to serve as references for future offline reinforcement learning research and its practical applications. By sharing comprehensive logs, researchers can readily access and utilize our results without having to re-run any of our experiments, ensuring that the results are replicable.

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

# A Additional Benchmark Information

## A.1 Offline

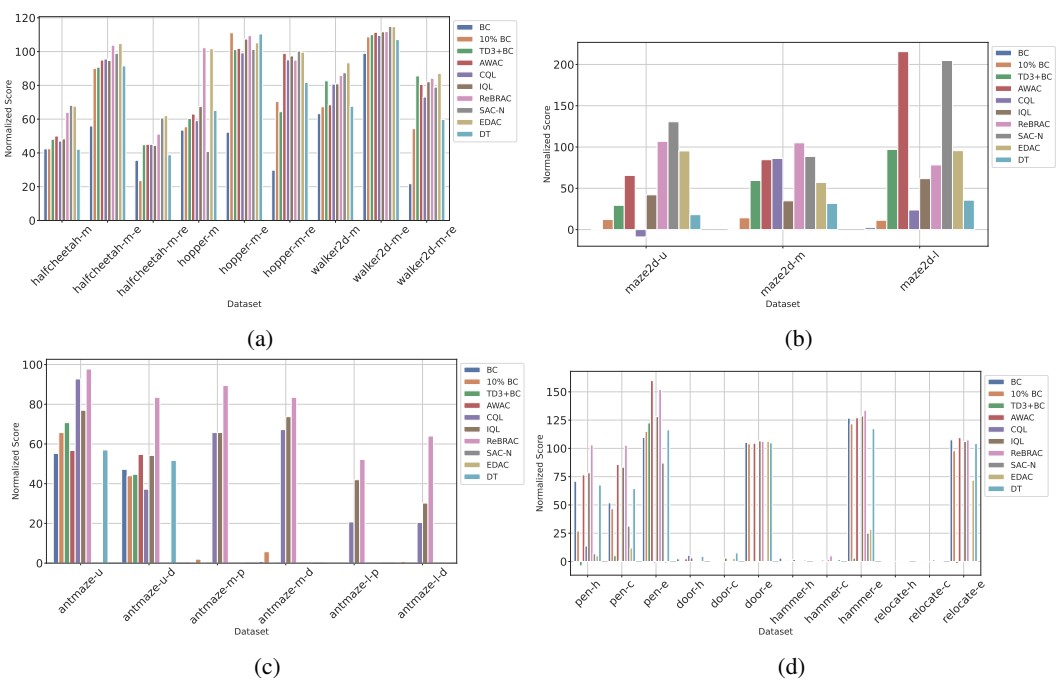

(a)

(b)

(c)

(d)

Figure 4: Graphical representation of the normalized performance of the last trained policy on D4RL averaged over 4 random seeds. (a) Gym-MuJoCo datasets. (b) Maze2d datasets (c) AntMaze datasets (d) Adroit datasets

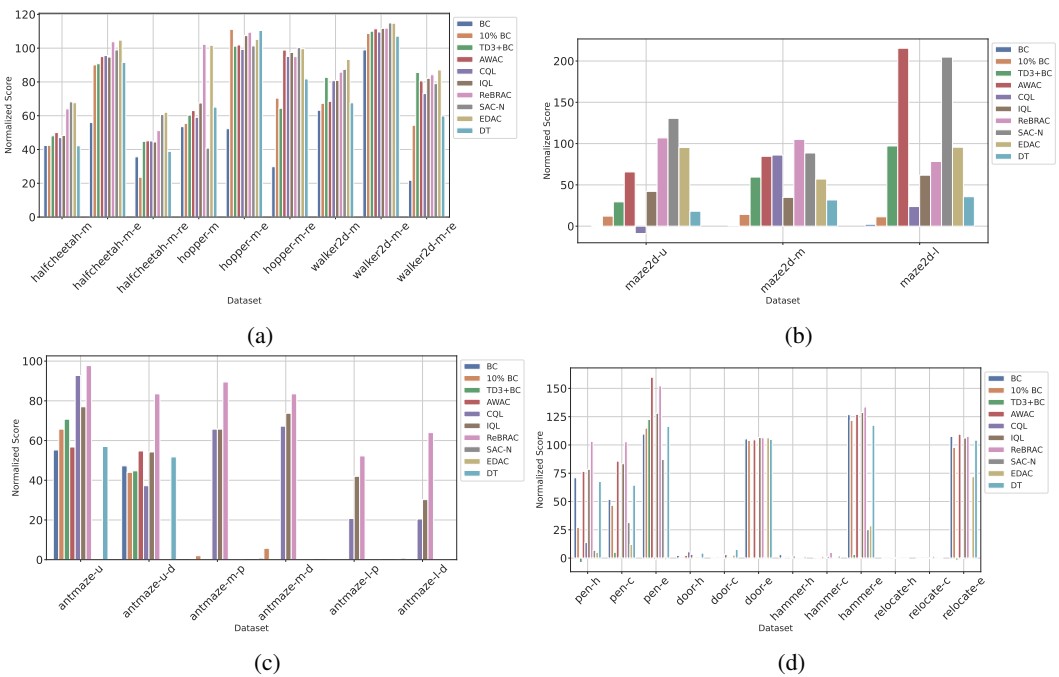

Figure 5: Graphical representation of the normalized performance of the best trained policy on D4RL averaged over 4 random seeds. (a) Gym-MuJoCo datasets. (b) Maze2d datasets (c) AntMaze datasets (d) Adroit datasets

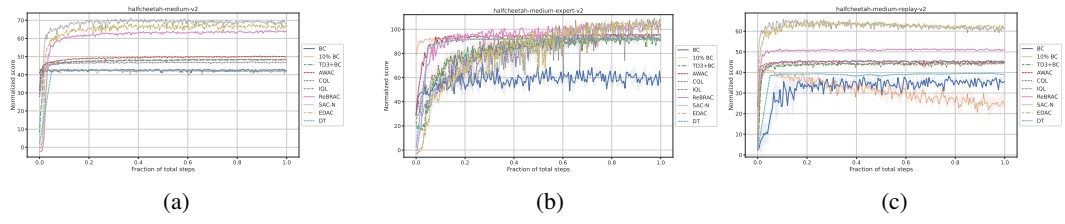

Figure 6: Training curves for HalfCheetah task.
(a) Medium dataset, (b) Medium-expert dataset, (c) Medium-replay dataset

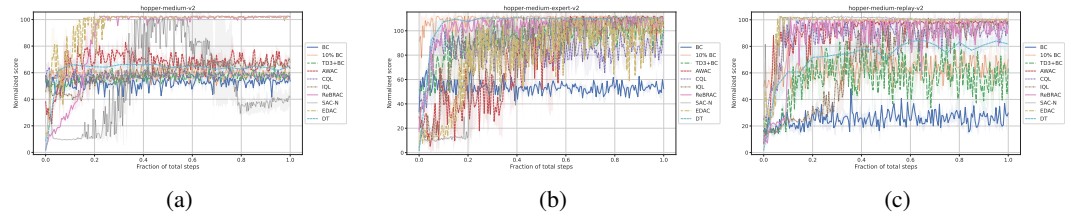

Figure 7: Training curves for Hopper task.
(a) Medium dataset, (b) Medium-expert dataset, (c) Medium-replay dataset

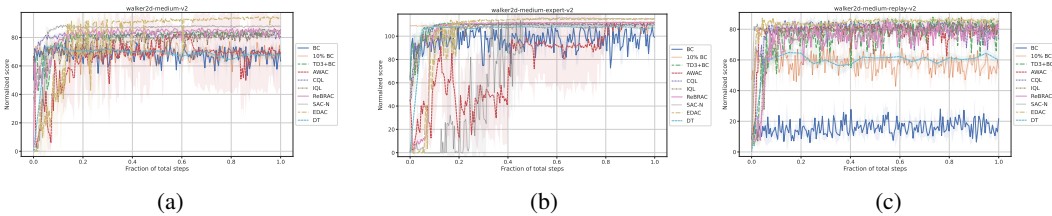

Figure 8: Training curves for Walker2d task.
(a) Medium dataset, (b) Medium-expert dataset, (c) Medium-replay dataset

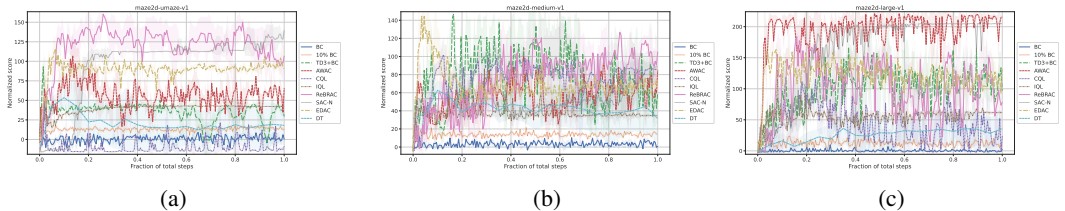

Figure 9: Training curves for Maze2d task.
(a) Medium dataset, (b) Medium-expert dataset, (c) Medium-replay dataset

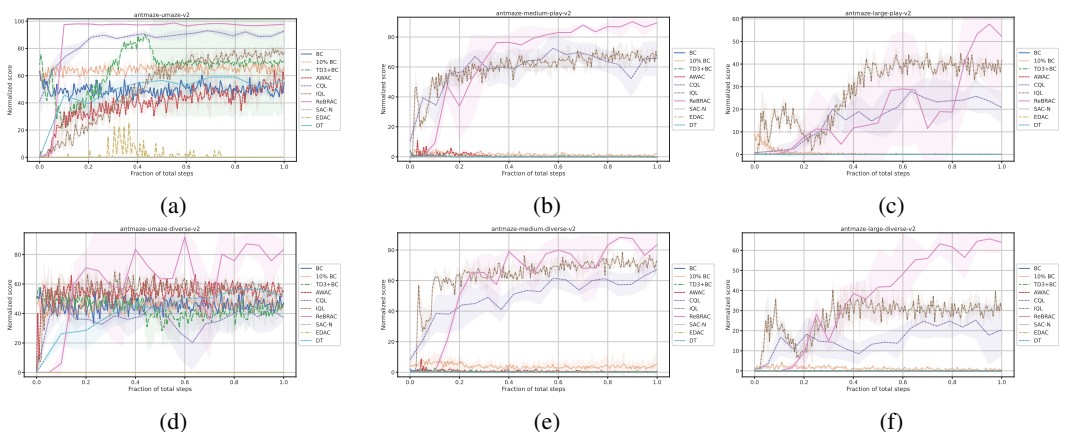

Figure 10: Training curves for AntMaze task.
(a) Umaze dataset, (b) Medium-play dataset, (c) Large-play dataset, (d) Umaze-diverse dataset, (e) Medium-diverse dataset, (f) Large-diverse dataset

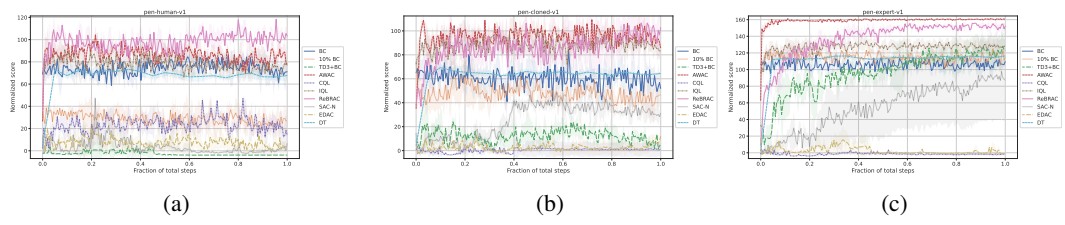

Figure 11: Training curves for Pen task.
(a) Human dataset, (b) Colned dataset, (c) Expert dataset

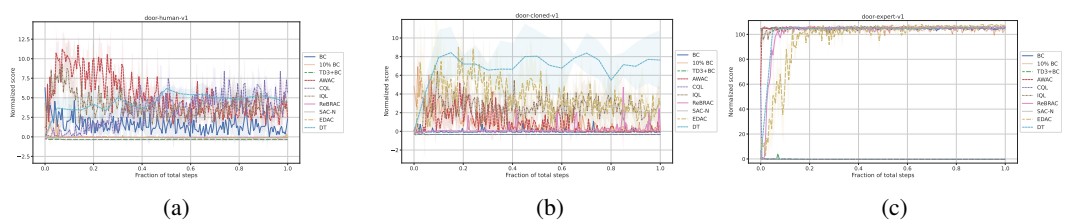

Figure 12: Training curves for Door task.
(a) Human dataset, (b) Colned dataset, (c) Expert dataset

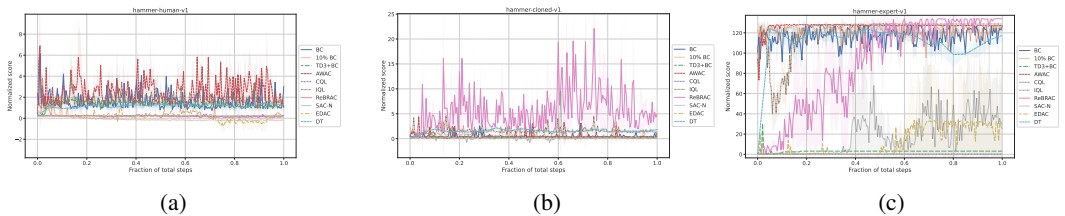

Figure 13: Training curves for Hammer task.
(a) Human dataset, (b) Colned dataset, (c) Expert dataset

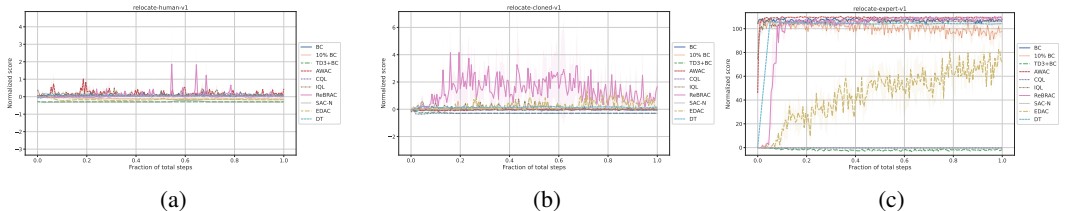

Figure 14: Training curves for Relocate task.
(a) Human dataset, (b) Colned dataset, (c) Expert dataset

## A.2 Offline-to-online

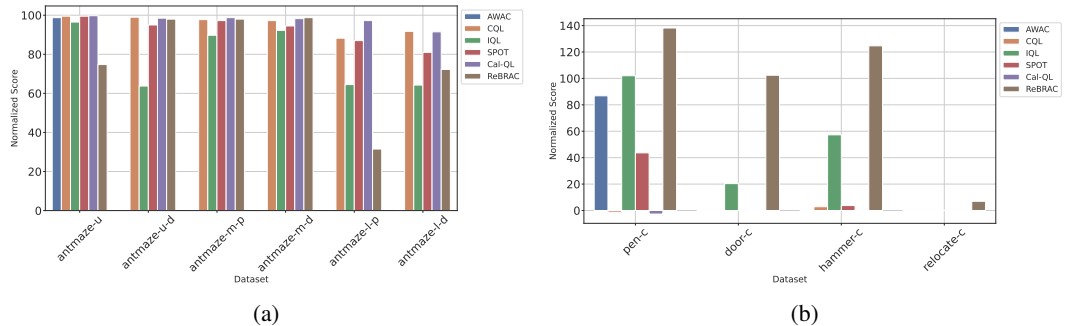

(a)                                  (b)

Figure 15: Graphical representation of the normalized performance of the last trained policy on D4RL after online tuning averaged over 4 random seeds.
(a) AntMaze datasets (b) Adroit datasets

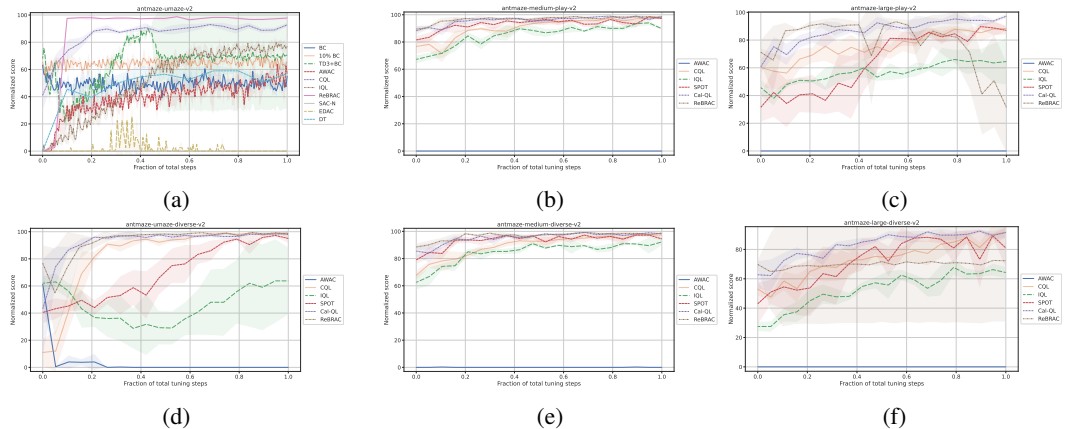

(a)                  (b)                  (c)

(d)                  (e)                  (f)

Figure 16: Training curves for AntMaze task during online tuning.
(a) Umaze dataset, (b) Medium-play dataset, (c) Large-play dataset, (d) Umaze-diverse dataset, (e) Medium-diverse dataset, (f) Large-diverse dataset

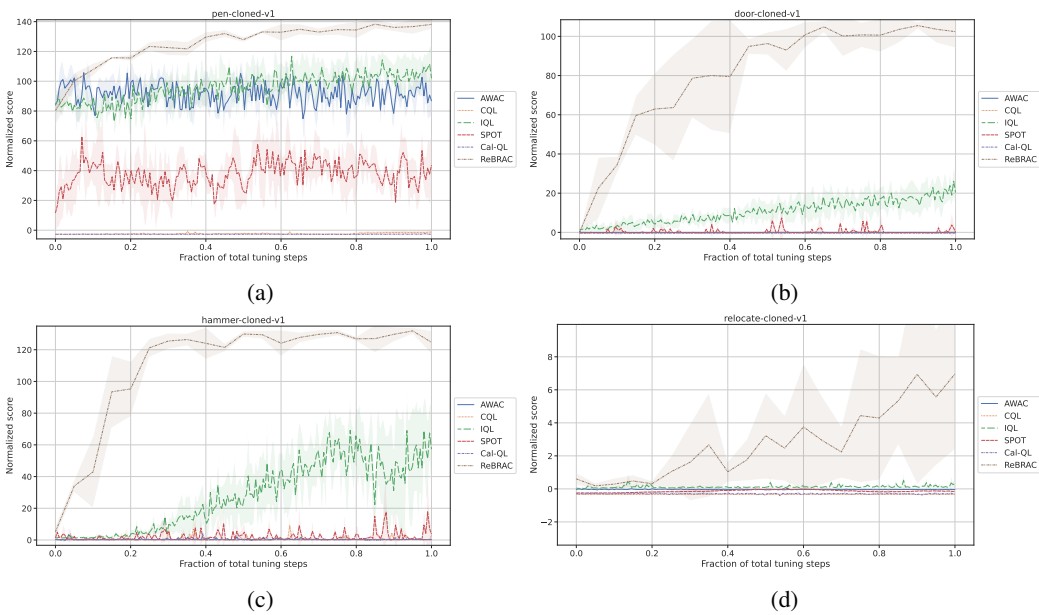

Figure 17: Training curves for Adroit Cloned task during online tuning.
(a) Pen, (b) Door, (c) Hammer, (d) Relocate

## B  Weights&Biases Tracking

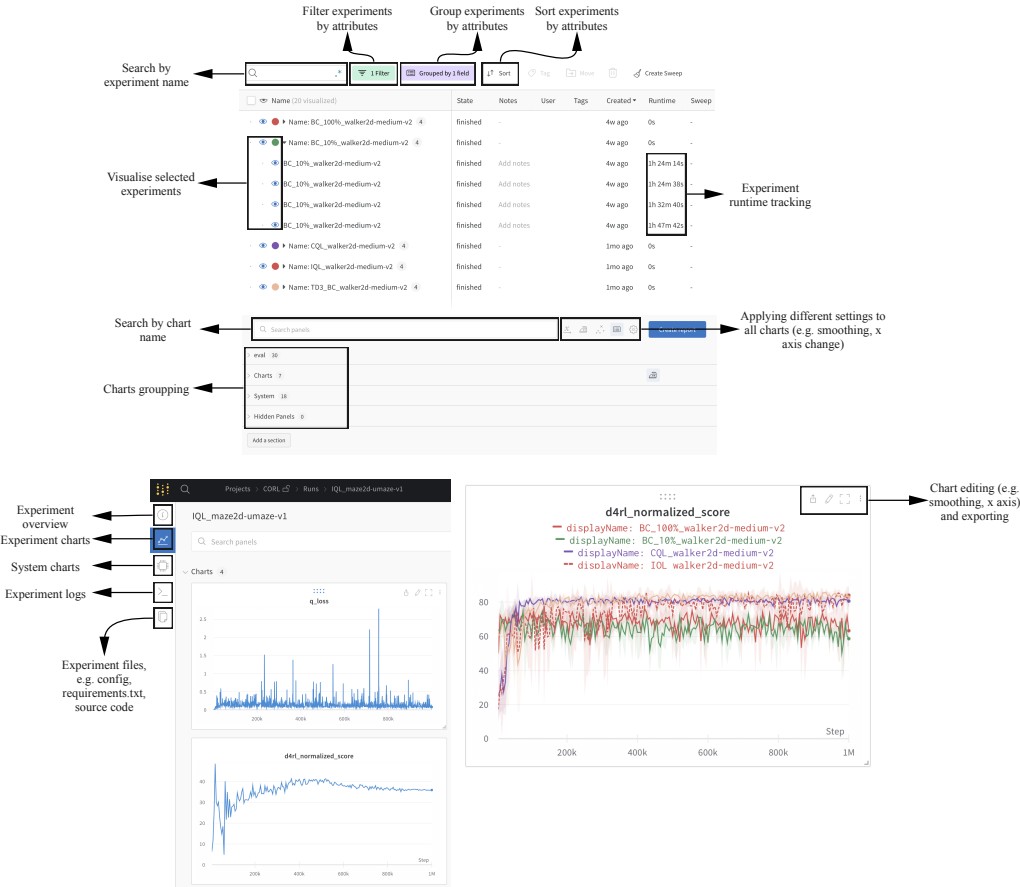

Figure 18: Screenshots of Weights&Biases experiment tracking interface.

## C  License

Our codebase is released under Apache License 2.0. The D4RL datasets (Fu et al., 2020) are released under Apache License 2.0.

## D    Experimental Details

We modify reward on AntMaze task by subtracting 1 from reward as it is done in previous works except CQL and Cal-QL, where (0, 1) are mapped into (-5, 5).

We used original implementation of TD3 + BC[14], SAC-$N$/EDAC[15], SPOT[16], ReBRAC[17] and custom implementations of IQL[18] and CQL/Cal-QL[19] as the basis for ours.

For most of the algorithms and datasets, we use default hyperparameters if available. Configuration files for every algorithm and environment are presented in our GitHub repository. Hyperparameters are also provided in subsection D.2.

All the experiments ran using V100 and A100 GPUs, which took approximately 5000 hours of compute in total.

### D.1    Number of update steps and evaluation rate

Following original work, SAC-$N$ and EDAC are trained for 3 million steps (except AntMaze, which is trained for 1 million steps) in order to obtain state-of-the-art performance and tested every 10000 steps. Decision Transformer (DT) training is splitted into datasets pass epochs. We train DT for 50 epochs on each dataset and evaluate every 5 epochs. All other algorithms are trained for 1 million steps and evaluated every 5000 steps (50000 for AntMaze). We evaluate every policy for 10 episodes on Gym-MuJoCo and Adroit tasks and for 100 for Maze2d and AntMaze tasks.

### D.2    Hyperparameters

Table 5: BC and BC-$N\%$ hyperparameters. † used for the best trajectories choice.

|  | Hyperparameter | Value |
|---|---|---|
| BC hyperparameters | Optimizer | Adam (Kingma & Ba, 2014) |
|  | Learning Rate | 3e-4 |
|  | Mini-batch size | 256 |
| Architecture | Policy hidden dim | 256 |
|  | Policy hidden layers | 2 |
|  | Policy activation function | ReLU |
| BC-$N\%$ hyperparameters | Ratio of best trajectories used | 0.1 |
|  | Discount factor† | 1.0 |
|  | Max trajectory length† | 1000 |

---

[14] https://github.com/sfujim/TD3_BC
[15] https://github.com/snu-mllab/EDAC
[16] https://github.com/thuml/SPOT
[17] https://github.com/tinkoff-ai/ReBRAC
[18] https://github.com/gwthomas/IQL-PyTorch
[19] https://github.com/young-geng/CQL

Table 6: TD3+BC hyperparameters.

| | Hyperparameter | Value |
|---|---|---|
| | Optimizer | Adam (Kingma & Ba, 2014) |
| | Critic learning rate | 3e-4 |
| | Actor learning rate | 3e-4 |
| | Mini-batch size | 256 |
| TD3 hyperparameters | Discount factor | 0.99 |
| | Target update rate | 5e-3 |
| | Policy noise | 0.2 |
| | Policy noise clipping | (-0.5, 0.5) |
| | Policy update frequency | 2 |
| | Critic hidden dim | 256 |
| | Critic hidden layers | 2 |
| Architecture | Critic activation function | ReLU |
| | Actor hidden dim | 256 |
| | Actor hidden layers | 2 |
| | Actor activation function | ReLU |
| TD3+BC hyperparameters | $\alpha$ | 2.5 |

Table 7: CQL and Cal-QL hyperparameters. Note: used hyperparameters are suboptimal on Adroit for the implementation we provide.

| | Hyperparameter | Value |
|---|---|---|
| | Optimizer | Adam (Kingma & Ba, 2014) |
| | Critic learning rate | 3e-4 |
| | Actor learning rate | 1e-4 |
| SAC hyperparameters | Mini-batch size | 256 |
| | Discount factor | 0.99 |
| | Target update rate | 5e-3 |
| | Target entropy | -1 · Action Dim |
| | Entropy in Q target | False |
| | Critic hidden dim | 256 |
| | Critic hidden layers | 5, AntMaze |
| | | 3, otherwise |
| Architecture | Critic activation function | ReLU |
| | Actor hidden dim | 256 |
| | Actor hidden layers | 3 |
| | Actor activation function | ReLU |
| | Lagrange | True, Maze2d and AntMaze |
| | | False, otherwise |
| | Offline $\alpha$ | 1.0, Adroit |
| | | 5.0, AntMaze |
| CQL hyperparameters | | 10.0, otherwise |
| | Lagrange gap | 5, Maze2d |
| | | 0.8, AntMaze |
| | Pre-training steps | 0 |
| | Num sampled actions (during eval) | 10 |
| | Num sampled actions (logsumexp) | 10 |
| | Mixing ratio | 0.5 |
| Cal-QL hyperparameters | Online $\alpha$ | 1.0, Adroit |
| | | 5.0, AntMaze |

Table 8: IQL hyperparameters.

| | Hyperparameter | Value |
|---|---|---|
| IQL hyperparameters | Optimizer | Adam (Kingma & Ba, 2014) |
| | Critic learning rate | 3e-4 |
| | Actor learning rate | 3e-4 |
| | Value learning rate | 3e-4 |
| | Mini-batch size | 256 |
| | Discount factor | 0.99 |
| | Target update rate | 5e-3 |
| | Learning rate decay | Cosine |
| | Deterministic policy | True, Hopper Medium and Medium-replay |
| | | False, otherwise |
| | $\beta$ | 6.0, Hopper Medium-expert |
| | | 10.0, AntMaze |
| | | 3.0, otherwise |
| | $\tau$ | 0.9, AntMaze |
| | | 0.5, Hopper Medium-expert |
| | | 0.7, otherwise |
| Architecture | Critic hidden dim | 256 |
| | Critic hidden layers | 2 |
| | Critic activation function | ReLU |
| | Actor hidden dim | 256 |
| | Actor hidden layers | 2 |
| | Actor activation function | ReLU |
| | Value hidden dim | 256 |
| | Value hidden layers | 2 |
| | Value activation function | ReLU |

Table 9: AWAC hyperparameters.

| | Hyperparameter | Value |
|---|---|---|
| AWAC hyperparameters | Optimizer | Adam (Kingma & Ba, 2014) |
| | Critic learning rate | 3e-4 |
| | Actor learning rate | 3e-4 |
| | Mini-batch size | 256 |
| | Discount factor | 0.99 |
| | Target update rate | 5e-3 |
| | $\lambda$ | 0.1, Maze2d, AntMaze |
| | | 0.3333, otherwise |
| Architecture | Critic hidden dim | 256 |
| | Critic hidden layers | 2 |
| | Critic activation function | ReLU |
| | Actor hidden dim | 256 |
| | Actor hidden layers | 2 |
| | Actor activation function | ReLU |

Table 10: SAC-$N$ and EDAC hyperparameters.

| | Hyperparameter | Value |
|---|---|---|
| SAC hyperparameters | Optimizer | Adam (Kingma & Ba, 2014) |
| | Critic learning rate | 3e-4 |
| | Actor learning rate | 3e-4 |
| | $\alpha$ learning rate | 3e-4 |
| | Mini-batch size | 256 |
| | Discount factor | 0.99 |
| | Target update rate | 5e-3 |
| | Target entropy | -1 · Action Dim |
| Architecture | Critic hidden dim | 256 |
| | Critic hidden layers | 3 |
| | Critic activation function | ReLU |
| | Actor hidden dim | 256 |
| | Actor hidden layers | 3 |
| | Actor activation function | ReLU |
| SAC-N hyperparameters | Number of critics | 10, HalfCheetah |
| | | 20, Walker2d |
| | | 25, AntMaze |
| | | 200, Hopper Medium-expert, Medium-replay |
| | | 500, Hopper Medium |
| EDAC hyperparameters | Number of critics | 10, HalfCheetah |
| | | 10, Walker2d, AntMaze |
| | | 50, Hopper |
| | $\mu$ | 5.0, HalfCheetah Medium-expert, Walker2d Medium-expert |
| | | 1.0, otherwise |

Table 11: DT hyperparameters.

|  | Hyperparameter | Value |
|---|---|---|
| DT hyperparameters | Optimizer | AdamW (Loshchilov & Hutter, 2017) |
|  | Batch size | 256, AntMaze |
|  |  | 4096, otherwise |
|  | Return-to-go conditioning | (12000, 6000), HalfCheetah |
|  |  | (3600, 1800), Hopper |
|  |  | (5000, 2500), Walker2d |
|  |  | (160, 80), Maze2d umaze |
|  |  | (280, 140), Maze2d medium and large |
|  |  | (1, 0.5), AntMaze |
|  |  | (3100, 1550), Pen |
|  |  | (2900, 1450), Door |
|  |  | (12800, 6400), Hammer |
|  |  | (4300, 2150), Relocate |
|  | Reward scale | 1.0, AntMaze |
|  |  | 0.001, otherwise |
|  | Dropout | 0.1 |
|  | Learning rate | 0.0008 |
|  | Adam betas | (0.9, 0.999) |
|  | Clip grad norm | 0.25 |
|  | Weight decay | 0.0003 |
|  | Total gradient steps | 100000 |
|  | Linear warmup steps | 10000 |
| Architecture | Number of layers | 3 |
|  | Number of attention heads | 1 |
|  | Embedding dimension | 128 |
|  | Activation function | GELU |

Table 12: SPOT hyperparameters.

|  | Hyperparameter | Value |
|---|---|---|
| VAE hyperparameters | Optimizer | Adam (Kingma & Ba, 2014) |
|  | Learning rate | 1e-3 |
|  | Mini-batch size | 256 |
|  | Number of iterations | $10^5$ |
|  | KL term weight | 0.5 |
| VAE architecture | Encoder hidden dim | 750 |
|  | Encoder layers | 3 |
|  | Latent dim | $2 \times$ action dim |
|  | Decoder hidden dim | 750 |
|  | Decoder layers | 3 |
| TD3 hyperparameters | Optimizer | Adam (Kingma & Ba, 2014) |
|  | Critic learning rate | 3e-4 |
|  | Actor learning rate | 1e-4 |
|  | Mini-batch size | 256 |
|  | Discount factor | 0.99 |
|  | Target update rate | 5e-3 |
|  | Policy noise | 0.2 |
|  | Policy noise clipping | (-0.5, 0.5) |
|  | Policy update frequency | 2 |
| Architecture | Critic hidden dim | 256 |
|  | Critic hidden layers | 2 |
|  | Critic activation function | ReLU |
|  | Actor hidden dim | 256 |
|  | Actor hidden layers | 2 |
|  | Actor activation function | ReLU |
| SPOT hyperparameters | $\lambda$ | 0.05, 0.1, 0.2, 0.5, 1.0, 2.0, AntMaze 1.0, Adroit |

