# OpenReview forum: "CORL: Research-oriented Deep Offline Reinforcement Learning Library"
_NeurIPS.cc/2023/Track/Datasets_and_Benchmarks — NeurIPS 2023 Datasets and Benchmarks Poster_

### Official Review · Reviewer_KZpC · 2023-06-30

**Rating:** 6
**Confidence:** 3
**Correctness:** Most descriptions about offline RL an…

**Strengths:**

- CORL enables accurate, quick, and reproducible implementation and empirical comparison of a range of offline and offline-online RL algorithms for the first time


- The paper is well-written and the arguments are easy to follow


- Figure 1 is useful to quickly understand the merits of CORL

**Additional Feedback:**

My suggestions are described in the opportunity for improvement section above.

**Clarity:**

The arguments, software characteristics, and experiment design are well-written and clear.

**Documentation:**

There seems to be no documentation website of the proposed library. For example, the famous VowpalWabbit package for contextual bandits has this rich documentation and tutorials: https://vowpalwabbit.org/index.html

**Ethics:**

No ethical concerns

**Limitations:**

Limitations are not explicitly described in the main text.

**Opportunities For Improvement:**

- It was not sure if it is possible to easily implement a new offline RL algorithm on top of CORL. In related research, we would often like to compare a newly proposed algorithm with existing ones, and it is important to clarify if CORL enables it or not. So, it would be more useful to show a demonstration of implementing and comparing some (mock) new algorithms with existing methods rather than merely showing a comparison of existing methods in Section 4.


- Values in the result tables (Tables 1, 2, 3) are not visible and interpretable (they are simply too small). So, the presentation can be refined.

- I was not sure of the point of Section 4. That is, Section 4 basically performs empirical comparison of a range of existing offline RL algorithms, but how does this section emphasize the strength of CORL against existing packages?

**Relation To Prior Work:**

Related software regarding online and offline RL are sufficiently summarized in Section 2.

**Summary And Contributions:**

The paper introduces CORL (Clean Offline Reinforcement Learning), an open-source library designed for offline and offline-to-online reinforcement learning. CORL offers single-file implementations of reinforcement learning algorithms, making them easier to understand and modify. The library also includes an experiment tracking tool for cloud-based logging of metrics, hyperparameters, and more. An added strength of CORL is its benchmarking on widely used D4RL datasets, providing a reference point for robust evaluations. Overall, CORL aims to aid newcomers in understanding offline RL and assist researchers in developing new algorithms without dealing with intricate abstraction levels.

---

> ### Author Response · Authors · 2023-08-19
>
> We thank the reviewer for the valuable comments and concerns. We tried to address them as following.
>
> > Values in the result tables (Tables 1, 2, 3) are not visible and interpretable (they are simply too small). So, the presentation can be refined.
> >
>
>  In the updated version of the paper, we tried to make the results more visual and accessible. We excluded BC (as 10% BC is a stronger baseline) and SAC-N (as EDAC is based on SAC-N and better) algorithms from `Table 1` and `Table 2`, and also reduced the reported precision in all tables to make more space. For clarity, we have highlighted the best scores for each dataset by algortim in the tables. Full comparison with all algorithms can now be found in the appendix. We hope this made the results more accessible.
>
> > There seems to be no documentation website of the proposed library. For example, the famous VowpalWabbit package for contextual bandits has this rich documentation and tutorials: https://vowpalwabbit.org/index.html
> >
>
> To address this, we made a detailed documentation website (in similar style to cleanrl) with more details about the implemented algorithms and logged metrics. It also covers how to reproduce our paper results, set up all dependencies, and how to run new experiments. Finally, we described what the process of adding new algorithms and contributions to our library should be.
>
> During the rebuttal period, we were able to describe the DT, SAC-N, CQL, and Cal-QL algorithms in detail, and we plan to significantly expand and improve the documentation in future releases, adding more algorithms and new details to those already described.
>
> > I was not sure of the point of Section 4. That is, Section 4 basically performs empirical comparison of a range of existing offline RL algorithms, but how does this section emphasize the strength of CORL against existing packages?
> >
>
> We have described the main differences between our library and others in the first sections. However, we want our library to be more than just a tool for researchers to help implement new algorithms. CORL's main mission is to make life easier for offline RL researchers. This includes fully public and accessible training logs for all algorithms, a consistent approach to comparing algorithms, and openness about the strengths and weaknesses of popular offline RL algorithms. For example, using CORL, a researcher can import Wandb logs into his project and focus on his new method, rather than wasting time re-implementing baselines. Moreover, because CORL provides highly detailed results that can be trusted (but also easily verified), the researcher can compare not only with the methods in CORL, but also with other work based on CORL. The way we see it, Section 4 presents CORL from this point of view.

---

> > ### Comment · Reviewer_KZpC · 2023-08-19
> >
> > Thank you for the additional efforts and clarifications, which solve most of my initial concerns. Therefore, I will raise my score to 6.

---

### Official Review · Reviewer_BnL5 · 2023-07-20
**Single-file implementations for offline-RL**

**Rating:** 8
**Confidence:** 4
**Correctness:** Yes.
**Clarity:** Yes.

**Strengths:**

Having an easy-to-understand codebase for offline RL is highly important to research development. As pointed out by the authors and earlier works such as [1], implementation details are highly relevant to the performance of RL algorithms. CORL makes it easy for researchers to understand the existence of these details and help produce correct reproductions and good science. The authors also shared valuable research engineering insights derived from their reproducible benchmark (which includes tracked experiments), which is helpful for future research.

[1] https://openreview.net/forum?id=r1etN1rtPB

**Additional Feedback:**

N/A

**Documentation:**

Reproducibility looks good, but documentation can be improved.

**Opportunities For Improvement:**

Consider investing further in the documentation. For example, "CQL is extremely sensitive to the choice of hyperparameters" is mentioned in the authors' paper, but nowhere mentioned in the repository. You should consider making documentation such as https://docs.cleanrl.dev/rl-algorithms/sac/#overview/ to document these tricky aspects of the algorithms.


Misc:
* The text in Figure 1 is barely readable. Consider making simplifications.
* The text in the tables and figures is generally pretty small
* A minor stylistic nit is maybe do not do the yellow callout boxes, which are generally used in textbooks.

**Relation To Prior Work:**

Yes.

**Summary And Contributions:**

This work presents CORL, an open-source library that provides single-file implementations of deep offline and offline-to-online reinforcement learning algorithms. The key contributions are as follows:

* **Variety of implemented algorithms (which are easy to understand)**: CORL provides minimalistic single-file implementations for 9 offline RL algorithms like BC, TD3 + BC, CQL, IQL, AWAC, SAC-N, EDAC, and DT. These single-file implementations have minimal dependencies and abstractions, which are easy to read and understand. This enables fast prototyping and debugging.
* **Benchmarked implementation**: the authors benchmarked the implementation extensively on the D4RL dataset to ensure quality and implementation details are done right.

---

> ### Author Response · Authors · 2023-08-19
>
> Thank you for the review and valuable comments.
>
> > Consider investing further in the documentation. For example, "CQL is extremely sensitive to the choice of hyperparameters" is mentioned in the authors' paper, but nowhere mentioned in the repository. You should consider making documentation such as https://docs.cleanrl.dev/rl-algorithms/sac/#overview/ to document these tricky aspects of the algorithms.
> >
>
> We agree that documentation is an essential part of making the library as clear and easy to use as possible for offline reinforcement learning practitioners and researchers. To address this, we made a detailed documentation website (in similar style to cleanrl) with more details about the implemented algorithms and logged metrics. It also covers how to reproduce our paper results, set up all dependencies, and how to run new experiments. Finally, we described what the process of adding new algorithms and contributions to our library should be.
>
> During the rebuttal period, we were able to describe the DT, SAC-N, CQL, and Cal-QL algorithms in detail, and we plan to significantly expand and improve the documentation in future releases, adding more algorithms and new details to those already described. You can see the new documentation [here](https://corl-team.github.io/CORL/).
>
> > The text in the tables and figures is generally pretty small
> >
>
> Thank you for feedback! In the updated version of the paper, we tried to make the results more visual and accessible. We excluded BC (as 10% BC is a stronger baseline) and SAC-N (as EDAC is based on SAC-N and better) algorithms from Table 1 and Table 2, and also reduced the reported precision in all tables to make more space. For clarity, we have highlighted the best scores for each dataset by algortim in the tables. Full comparison with all algorithms can now be found in the appendix. We hope this made the results more accessible.
>
> > A minor stylistic nit is maybe do not do the yellow callout boxes, which are generally used in textbooks.
> >
>
> Since our work is almost entirely empirical, we feel that boxes like this allow us to summarize the main conclusions from our experience and multiple huge tables in a concise and succinct manner and a number of recent works do the same [1, 2, 3].
>
> References:
>
> 1. Lu, C., Ball, P. J., Rudner, T. G., Parker-Holder, J., Osborne, M. A., & Teh, Y. W. (2022). Challenges and opportunities in offline reinforcement learning from visual observations. *arXiv preprint arXiv:2206.04779*.
> 2. Kumar, A., Singh, A., Tian, S., Finn, C., & Levine, S. (2021). A workflow for offline model-free robotic reinforcement learning. *arXiv preprint arXiv:2109.10813*.
> 3. Kumar, A., Agarwal, R., Geng, X., Tucker, G., & Levine, S. (2022). Offline q-learning on diverse multi-task data both scales and generalizes. *arXiv preprint arXiv:2211.15144*.

---

> > ### Comment · Reviewer_BnL5 · 2023-08-25
> > **Response to the authors**
> >
> > I think the authors for their response. Given the improvement of the new documentation site, I have updated my ratings accordingly.

---

### Official Review · Reviewer_ueDk · 2023-07-20
**Review of the paper**

**Rating:** 7
**Confidence:** 4
**Correctness:** Yes

**Strengths:**

(1) The single-file design exposes algorithmic details for easier understanding, debugging, and modification compared to larger modular codebases.

(2) Lack of abstractions and independency between online/offline algorithm variants reduces implementation overhead and grants flexibility.

(3) Built-in benchmarking on D4RL datasets provides reliable baseline results to aid evaluation and avoid reimplementing algorithms.

**Additional Feedback:**

How does CORL compare with offline RL libraries like d3rlpy in terms of usability and performance?

**Clarity:**

The writing and the presentation could be improved. For example, we can hardly read anything from Figure 1. The authors could also highlight some results in the tables to facilitate reading.

**Documentation:**

Yes

**Limitations:**

Yes

**Opportunities For Improvement:**

(1) The simplicity comes at the cost of code duplication between similar algorithms and lack of modular reusability.

(2) There is limited comparison to prior offline RL libraries like d3rlpy on unique capabilities provided.

(3) More analysis could be provided on associations between algorithms and environment types based on the benchmarking.

**Relation To Prior Work:**

Yes

**Summary And Contributions:**

This paper presents CORL, an open-source library for deep offline and offline-to-online reinforcement learning. The key contribution is minimalistic, single-file implementations of RL algorithms to facilitate understanding and modification. CORL emphasizes isolating algorithm details, lightweight dependencies, and built-in benchmarking on D4RL tasks. Extensive experiments characterize algorithm performance and highlight insights like the sensitivity of CQL hyperparameters. Overall, CORL seems like a useful library for understanding RL algorithms and rapidly prototyping new ideas, especially for offline RL research.

---

> ### Author Response · Authors · 2023-08-19
>
> We thank the reviewer for the valuable comments and concerns.
>
> > The writing and the presentation could be improved. For example, we can hardly read anything from Figure 1. The authors could also highlight some results in the tables to facilitate reading.
> >
>
> Thank you for your feedback! In the updated version of the paper, we tried to make the results more visual and accessible. We excluded BC (as 10% BC is a stronger baseline) and SAC-N (as EDAC is based on SAC-N and better) algorithms from `Table 1` and `Table 2`, and also reduced the reported precision in all tables to make more space. For clarity, we have highlighted the best scores for each dataset by algorithm in the tables. Full comparison with all algorithms can now be found in the appendix. We hope this made the results more accessible.
>
> > How does CORL compare with offline RL libraries like d3rlpy in terms of usability and performance?
> >
>
> As we argue in our paper, the main advantage of our library is the complete transparency of the algorithms implementations, the lack of modularity, and the ability to fully control every single component. This allows one to quickly implement new algorithms based on existing ones, or to change specific details for new experiments. Whereas, in order to change something in d3rlpy, you have to go a long way. For example, the [`CQL`](https://github.com/takuseno/d3rlpy/blob/4ba297fc6cd62201f7cd4edb7759138182e4ce04/d3rlpy/algos/qlearning/cql.py#L135) algorithm inherits from `CQLImpl`, which in turn inherits from `SACImpl`, which in turn inherits from `DDPGBaseImpl`, which in turn inherits from `QLearningAlgoImplBase`. CQL, SAC, DDPG are all different algorithms that require different tricks and methods to make them work, so this approach effectively makes d3rlpy-based research efforts infeasible. While implementations in d3rlpy may be more optimized and faster in some specific cases, we emphasize a rather more minimalistic approach, more suited to the needs of researchers.

---

> > ### Comment · Reviewer_ueDk · 2023-08-30
> >
> > Thanks for your response. I have increased my score.

---

### Official Review · Reviewer_JMRf · 2023-07-21
**A useful single-file-implementation codebase for Offline RL algorithms, but still has room for improvement**

**Rating:** 7
**Confidence:** 4
**Correctness:** To the best of my knowledge, the clai…

**Strengths:**

* Every algorithm is isolated into a single file, making it much easier to both understand the code as well as modify it for novel methods.
* YAML config files are used to adjust and keep track of hyperparameters.
* Experiment tracking with W&B is integrated.
* Most results of the implementations are on par with the original implementations or other offline RL codebases.

**Additional Feedback:**

Overall, I'm happy to see the existence of CORL for the offline RL community. The single-file implementation does indeed help a lot with code readability, though I would prefer not going to the extreme and still keeping some modularness in the code (see Opportunities for Improvement #1). A few more minor details:

1. One of the first few lines of the paper's Github repo writes that CORL is "Benchmarked Implementation for N algorithms". Is the "N" here intentional? I feel like it would be better to directly write out the number, which I counted to be 13.
2. There are a lot of unused variables in the code, for example `alpha_prime` in L907 of `finetune/cal_ql.py`. I don't if this was intentional, and if not these variables can be replaced by the `_` symbol.
3. In `offline/rebrac.py`, the `train()` function in all the other files is named `main()` instead. I would suggest keeping the naming consistent and changing it to `train()` as well.

My main concerns are the issues mentioned in the Limitations section, especially issue #1, so I'm currently giving a 5. If the authors can address these issues and fix the other problems I pointed out, I'd be happy to raise my score and accept this paper.

**Clarity:**

In general, the paper is clearly written. I found a typo on line 29 (should be "Weights&Biases", not "Weighs&Biases"). On line 114, I'm not sure what "max archived score" exactly meant, is it basically the "best" score used in the rest of the paper or does it have other meanings?

**Documentation:**

There are sufficient details to support reproducibility of the results. Overall I found the repo's instructions easy to follow, and was able to install & run CORL in a short period of time.

However, it would be even better if the authors published CORL to `pypi` so it can be directly `pip`-installed, or if a `setup.py` was simply added so that one line of `pip install git+...` could be used for installation.

I also suggest adding one line of running-code instruction to the repo's README, like the `python dt.py --config=cfg/dt-hopper.yaml --logdir=logs/dt-hopper --num-epochs=50` line in Figure 1.

**Ethics:**

I do not see any ethical concerns in this work.

**Limitations:**

1. I found different training lengths for different seeds in the published results. I downloaded the released `pickle` file of results for the offline algorithms, and found these algorithms and datasets had different number of score evaluations (training lenghts) for different seeds. Does this mean these seeds were not run toghether? It would also affect both the best and final scores. Here are the training lengths of 4 seeds that were not consistent, with corresponding algorithms and datasets:
```
AWAC antmaze-umaze-v2 1000 1000 1001 1001
AWAC maze2d-large-v1 1000 1000 1000 200
AWAC maze2d-medium-v1 1000 1000 1000 200
AWAC maze2d-umaze-v1 1000 1000 1000 200
AWAC walker2d-medium-expert-v2 1000 1000 1000 200
AWAC walker2d-medium-replay-v2 1000 1000 1000 200
AWAC walker2d-medium-v2 1000 1000 1000 200
AWAC hopper-medium-expert-v2 1000 1000 1000 200
AWAC hopper-medium-replay-v2 1000 1000 1000 200
AWAC hopper-medium-v2 1000 1000 1000 200
AWAC halfcheetah-medium-expert-v2 1000 1000 1000 200
AWAC halfcheetah-medium-replay-v2 1000 1000 1000 200
AWAC halfcheetah-medium-v2 1000 1000 1000 200
SAC-N relocate-human-v1 40 40 40 601
SAC-N relocate-expert-v1 601 601 601 42
EDAC relocate-human-v1 44 601 601 601
```
2. The codebase is not updated to the latest version of libraries. For example it still uses [`gym`](https://github.com/openai/gym), which is already deprecated and replaced by [`gymnasium`](https://github.com/Farama-Foundation/Gymnasium);  [`mujoco-py`](https://github.com/openai/mujoco-py/) is now replaced by [`mujoco`](https://github.com/deepmind/mujoco/blob/main/python/README.md) (which is also much easier to install); [`torch`](https://pytorch.org/) has updated to 2.0.1. I was trying to install CORL in a Python 3.11 environment and it failed due to some of the libraries requiring a Python version earlier than 3.11. I'm not asking for the libraries to be strictly the latest versions, but at least deprecated versions should be replaced.

**Opportunities For Improvement:**

1. Even though the non-modular style helps the files to be isolated and easier to read, there are also many components of basic RL code (e.g., the replay buffer) that are inevitably repeated for every single file. I understand that the authors want every file to run on its own, but I think that a limited number of abstraction layers (for example, only 2 or 3 layers of subclasses) would strike a good balance between readability and code efficiency.
2. The codebase uses `pyrallis` (100+ stars) for config. According to my knowledge, `hydra` (7k+ stars) is more commonly used for ML configs and logging, and the config in this repo does not use non-standard types.

**Relation To Prior Work:**

There are two prior works that are mainly related to this paper. The first one is [`d3rlpy`](https://github.com/takuseno/d3rlpy) which is also an offline RL codebase; the second one is [`CleanRL`](https://github.com/vwxyzjn/cleanrl) which takes a similar approach on building a non-module RL library. CORL is different from [`d3rlpy`](https://github.com/takuseno/d3rlpy) as it is non-modular, while [`CleanRL`](https://github.com/vwxyzjn/cleanrl) only focuses on implementations of online RL algorithms. CORL also minimizes the requirements of external dependencies, compared to both prior works.

**Summary And Contributions:**

The Clean Offline Reinforcement Learning (CORL) paper introduces a novel offline RL baseline codebase, designed to implement cutting-edge offline and offline-to-online RL algorithms in *single* files. Unlike traditional approaches, CORL eliminates excessive abstraction layers, resulting in a codebase that is highly readable, comprehensible, and readily adaptable for new methods. CORL seamlessly integrates with the standard [W&B](https://wandb.ai/) experiment tracking tool, providing researchers with a convenient means of directly monitoring their results. Moreover, the paper reinforces the codebase's credibility by openly sharing the results, hyperparameters, and training logs, ensuring transparency and demonstrating the reliability and performance of its implementations. Overall, CORL presents a promising and accessible solution for offline RL experimentation.

---

> ### Author Response · Authors · 2023-08-19
>
> We would like to thank you for your time reviewing our paper results and valuable comments. We will try to address them below.
>
> > I found different training lengths for different seeds in the published results. I downloaded the released pickle file of results for the offline algorithms, and found these algorithms and datasets had different number of score evaluations (training lenghts) for different seeds. Does this mean these seeds were not run toghether? It would also affect both the best and final scores
>
> Thanks for finding the bug! Indeed, after some investigation, it turned out that those particular seeds were incorrect. We have recomputed them for the AWAC algorithm and updated the scores in the paper and README tables ([PR with fix](https://github.com/corl-team/CORL/commit/44a9da4e3bcedce2e788cf57abbf028a26647b66)). Unfortunately SAC-N and EDAC are diverging on these seeds, causing some runs to crash prematurely, so the score is estimated on the cropped minimum length for all seeds, which probably underestimates the final score. It will take us some time to find new hyperparameters for these datasets, and since it's only a three of them, we'll try to fix this for the next major release.
>
> > The codebase is not updated to the latest version of libraries. I'm not asking for the libraries to be strictly the latest versions, but at least deprecated versions should be replaced.
>
> While we understand the possible frustration of installing dependencies currently, there is nothing we can do about it at the moment, since CORL relies almost entirely on dependencies of D4RL, which is still the standard benchmark in the offline rl community. However, D4RL stopped being updated long ago and is effectively unsupported, using older versions of many libraries (including mujoco-py and gym). We know this is an issue and have recently been actively working with the [Farama-Foundation/Minari](https://github.com/Farama-Foundation/Minari) team to add Minari support to CORL ([ongoing PR](https://github.com/corl-team/CORL/pull/3)). This will allow us to significantly update dependencies and simplify installation, and give users access to many new datasets out of the box!
>
> > I found a typo on line 29 (should be "Weights&Biases", not "Weighs&Biases"). On line 114, I'm not sure what "max archived score" exactly meant.
>
> Thanks, we fixed them in the updated version of the paper.
>
> > There are a lot of unused variables in the code, for example `alpha_prime` in L907 of `finetune/cal_ql.py`. I don't if this was intentional, and if not these variables can be replaced by the `_`symbol. In `offline/rebrac.py`, the `train()` function in all the other files is named `main()` instead. I would suggest keeping the naming consistent and changing it to `train()` as well.
>
> Thanks, these are indeed unused variables and inconsistent naming. We fixed it [here](https://github.com/corl-team/CORL/commit/2a30bc20a24265972dfec7688f8c33209f22544e).
>
> > One of the first few lines of the paper's Github repo writes that CORL is "Benchmarked Implementation for N algorithms". Is the "N" here intentional? I feel like it would be better to directly write out the number, which I counted to be 13.
>
> Indeed, initially it was intentional. Now it does not reflect the strengths of our library, so we changed to an explicit number in the [README](https://github.com/corl-team/CORL/blob/main/README.md) and [documentation](https://corl-team.github.io/CORL/).
>
> > I also suggest adding one line of running-code instruction to the repo's README, like the `python dt.py --config=cfg/dt-hopper.yaml --logdir=logs/dt-hopper --num-epochs=50` line in Figure 1.
>
> Thanks! We added a whole [new page](https://corl-team.github.io/CORL/get-started/usage/) with running code and experiments instructions on our new documentation website.
>
> > The codebase uses `pyrallis` (100+ stars) for config. According to my knowledge, `hydra` (7k+ stars) is more commonly used for ML configs and logging, and the config in this repo does not use non-standard types.
>
> While Hydra is more popular, it is much more complex and sophisticated, which doesn't quite accommodate the minimalism of our library. Whilst pyrallis is very simple and does exactly what it needs to do.
>
> > However, it would be even better if the authors published CORL to `pypi` so it can be directly `pip`-installed, or if a `setup.py` was simply added so that one line of `pip install git+...` could be used for installation.
>
> Indeed, that would be simpler. However, it seems that neither [setup.py](http://setup.py/) nor pyproject.toml support the use of `--extra-index-url` or `--find-links`, which is what our dependencies use in the `requirements.txt`. We would welcome advice if we are missing something. For now we added a warning to the README and documentation, stating that CORL is not meant to be imported.

---

> > ### Comment · Reviewer_JMRf · 2023-08-20
> >
> > I appreciate the extra dedication and explanations provided, which have effectively addressed the majority of my initial worries. As a result, I'll now raise my score to 7.

---

### Official Review · Reviewer_aaf5 · 2023-07-24

**Rating:** 7
**Confidence:** 4
**Correctness:** The claim of the paper is sound.
**Clarity:** The paper is well written.

**Strengths:**

The strengths of the paper include:

1) The authors provide clean and modular implementations of common offline RL algorithms. The code is well-organized and easy to understand.

2) The codebase is self-contained and does not require any additional dependencies on third-party libraries. This makes it easy to install and use.

3) The authors provide single configuration files for all D4RL tasks. This makes it easy for users to track all hyperparameters without having to jump to different places.

4) The authors provide comprehensive benchmark results and analysis. This information is valuable for researchers and practitioners who are interested in offline RL.

5) The codebase is integrated with Weights and Biases. This makes it easy to track experiments and share results.

**Additional Feedback:**

N/A

**Documentation:**

The documentation of the code can be improved.

**Opportunities For Improvement:**

1) The authors should improve the code documentation. Currently, the code is almost "comment-free," which makes it difficult for first-time users to understand what is happening within the algorithms. Adding comments and documentation would make the code more accessible and easier to use.

2) The authors should compare their implementations with other implementations of the same offline RL algorithms on selected tasks. This would (a) demonstrate the quality of the provided code and (b) help to identify any bugs or regressions in previously reported results.




**Relation To Prior Work:**

Yes.

**Summary And Contributions:**

In this paper, the authors present an offline deep reinforcement learning (RL) benchmark. They reimplement many well-known offline RL algorithms in a clean and modular fashion. They also provide single-file experiment configurations to simplify the process of testing all D4RL tasks. Furthermore, the authors report detailed findings from the benchmark, comparing the strengths and weaknesses of individual algorithms on various tasks.

---

> ### Author Response · Authors · 2023-08-19
> **Official Comment by Authors**
>
> Thank you for the review and for emphasizing our library's strengths. In terms of opportunities for improvement, we have tried to address the issues you note.
>
> > The authors should improve the code documentation. Currently, the code is almost "comment-free," which makes it difficult for first-time users to understand what is happening within the algorithms. Adding comments and documentation would make the code more accessible and easier to use.
> >
>
> We agree that documentation and comments in the code are an essential part of making the library as clear and easy to use as possible for offline reinforcement learning practitioners and researchers.
>
> To address this, we have increased the number of comments in the code, especially describing in detail the purpose of all possible hyperparameters in the training configs. Additionally, we also made a detailed documentation website with more details about the implemented algorithms and logged metrics. It also covers how to reproduce our paper results, set up all dependencies, and how to run new experiments. Finally, we described what the process of adding new algorithms and contributions to our library should be.
>
> During the rebuttal period, we were able to describe the DT, SAC-N, CQL, and Cal-QL algorithms in detail, and we plan to significantly expand and improve the documentation in future releases, adding more algorithms and new details to those already described. You can see the new documentation [here](https://corl-team.github.io/CORL/), and the PR with new code comments [here](https://github.com/corl-team/CORL/commit/23f8b2e46abf5aef73db268cd1c5a42eb8ef45dc).
>
> > The authors should compare their implementations with other implementations of the same offline RL algorithms on selected tasks. This would (a) demonstrate the quality of the provided code and (b) help to identify any bugs or regressions in previously reported results.
> >
>
> Although we acknowledge that comparing with another implementation can be beneficial, we believe that we have already taken the appropriate approach here. Numerous official and unofficial implementations exist for each method; however, most of them are not community-validated and do not offer open metrics and results for all datasets considered in our work. Therefore, comparing to something other than an official implementation would, in our opinion, be arbitrary and not very productive. In our work, we specifically compare with the results of the original implementation. As you may have noticed, in each of our public wandb reports, the reference scores of the original implementation are written next to the training logs of our implementation. This way everyone can verify that we have correctly reproduced the algorithm.

---

### Author Response · Authors · 2023-08-19

We thank all reviewers for their valuable comments and their time. Here we want to summarise all the new changes we made during the rebuttal period.

First of all, we have revised the paper, making the tables more visible and accessible, as well as correcting typos. We have also added one new method, ReBRAC, updating all relevant sections. The most important update is the introduction of a documentation site where we have described in detail the important aspects of using our library. During the rebuttal period, we were able to describe the DT, SAC-N, CQL, and Cal-QL algorithms in detail, and we plan to significantly expand and improve the documentation in future releases, adding more algorithms and new details to those already described.

Finally, due to unfortunate circumstances, CORL library moved to the [new fork](https://github.com/corl-team/CORL) and all further development and support will be there from this point onwards.

---

### Decision · Program_Chairs · 2023-09-22

**Decision:**

Accept (Poster)

**Comment:**

**Decision: Recommend Accept**

This paper presents an new open source library of benchmark offline and offline-to-online RL algorithms. Of note is that such implementations all are single-file, often in 400-600 hundred lines of code.  This design decision makes it easy for researchers to understand algorithms, as well as fork them to develop new ones. The library is benchmarked comprehensively on D4RL.

**Reviewer Commendations:**
* Several reviewers praised the code for being clean, modular, and self-contained with minimal dependencies.
* Simliarly the code was also praised for already being 'ready' for practical experimental work, for instance with Weights & Biases integration, and useful yaml files for D4RL.
* Several reviewers praised the variety of algorithms are implemented and their benchmarking on D4RL

**Reviewer Criticisims:**
* Several reviewers noted that there are probably _some_ areas of code reuse & abstractions could be useful - eg the replay buffer.
* Some also pointed out a similarity to prior work in d3rlpy or CleanRL, though these were not criticisms per se.

**AC View**

I agree with the majority of the reviewers that these clean, single-file implementations and the supporting benchmarking around them are likely to be highly useful to the Offline RL community, and recommend this for acceptance.